

# Evaluating FAIR Digital Object and Linked Data as distributed object systems

Stian Soiland-Reyes[1,2], Carole Goble[1] and Paul Groth[2]

[1] Department of Computer Science, The University of Manchester, Manchester, UK
[2] Informatics Institute, University of Amsterdam, Amsterdam, Netherlands

## ABSTRACT

FAIR Digital Object (FDO) is an emerging concept that is highlighted by European Open Science Cloud (EOSC) as a potential candidate for building an ecosystem of machine-actionable research outputs. In this work we systematically evaluate FDO and its implementations as a global distributed object system, by using five different conceptual frameworks that cover interoperability, middleware, FAIR principles, EOSC requirements and FDO guidelines themself. We compare the FDO approach with established Linked Data practices and the existing Web architecture, and provide a brief history of the Semantic Web while discussing why these technologies may have been difficult to adopt for FDO purposes. We conclude with recommendations for both Linked Data and FDO communities to further their adaptation and alignment.

*An earlier version of this article was published as a preprint (Soiland-Reyes, Goble & Groth, 2023), which is embedded in the corresponding RO-Crate for this article (Soiland-Reyes, 2023a) and the PhD thesis (Soiland-Reyes, 2023b).*

# INTRODUCTION

The FAIR principles (Wilkinson et al., 2016) encourage sharing of scientific data with machine-readable metadata and the use of interoperable formats, and are being adapted by a wide range of research infrastructures. They have been recognised by the research community and policy makers as a goal to strive for European Commission (2016). In particular, the European Open Science Cloud (EOSC) has promoted adaptation of FAIR data sharing of data resources across electronic research infrastructures (Mons et al., 2017). The EOSC Interoperability Framework (Corcho et al., 2021) puts particular emphasis on how interoperability can be achieved technically, semantically, organisationally, and legally—laying out a vision of how data, publication, software and services can work together to form an ecosystem of digital objects that are extensively described. Such descriptions for interoperability connect a range of information—from protocols and presentations, to hardware designs and scientific workflows, including extensive metadata of the information itself.

Specifically, the EOSC Interoperability framework highlights the emerging FAIR Digital Object (FDO) concept (Schultes & Wittenburg, 2019) as a possible foundation for building a semantically interoperable ecosystem to fully realise the FAIR principles beyond

Corresponding author
Stian Soiland-Reyes,
soiland-reyes@manchester.ac.uk

individual repositories and infrastructures. The FDO approach has great potential, as it proposes strong requirements for identifiers, types, access and formalises interactive operations on objects.

In other discourse, Linked Data (*Bizer, Heath & Berners-Lee, 2009*) has been seen as an established set of principles based on Semantic Web technologies that can achieve the vision of the FAIR principles (*Bonino da Silva Santos et al., 2016*; *Hasnain & Rebholz-Schuhmann, 2018*). Yet regular researchers and developers of emerging platforms for computation and data management are reluctant to adapt such a "FAIR Linked Data approach" fully (*Verborgh & Vander Sande, 2020*), opting instead for custom in-house models and JSON-derived formats from RESTful Web services (*Meroño-Peñuela, Lisena & Martínez-Ortiz, 2021a*; *Neumann, Laranjeiro & Bernardino, 2021*). While such focus on simplicity allows for rapid development and highly specialised services, it raises wider concerns about interoperability (*Turcoane, 2014*; *Wilkinson et al., 2022a*).

One challenge that may, perhaps counter-intuitively, steer developers towards a not-invented-here mentality (*Stefi, 2015*; *Stefi & Hess, 2015*) when exposing their data on the Web is the heterogeneity and apparent complexity of Semantic Web approaches themselves (*Meroño-Peñuela, Lisena & Martínez-Ortiz, 2021b*).

These approaches—FDO and Linked Data—thus, form two of the major avenues for allowing developers and the wider research community to achieve the goal of FAIR data. Given their importance, in this article, we aim to compare FDO with Linked Data and the Web architecture in the context of the discourse around FAIR data.

Concretely, the contribution of this article is **a systematic comparison between FDO and Linked Data using five different conceptual frameworks** that capture different perspectives on interoperability and readiness for implementation.

The rest of this article is organised as follows: We begin with a background primer on FDO and Linked Data to provide a foundation for the work. In the Method section, we introduce the conceptual frameworks we use for comparison. Subsequently, in the Results section, we systematically step through the outcomes of applying these frameworks to both FDO and Linked Data. For each framework, we derive key observations. We end with a discussion of these results and their implications for both approaches and conclude.

## BACKGROUND AND RELATED WORK

In the following, we discuss the related work with respect to FAIR Digital Objects and Linked Data. We do so by looking through the lens of development of these technologies over time, including future directions.

### FAIR digital object

The concept of **FAIR Digital Objects** (*Schultes & Wittenburg, 2019*) has been introduced as a way to expose research data as active objects that conform to the FAIR principles (*Wilkinson et al., 2016*). This builds on the *Digital Object* (DO) concept (*Kahn & Wilensky, 2006*), first introduced by *Kahn & Wilensky (1995)* as a system of *repositories* containing *digital objects* identified by *handles* and described by *metadata* which may have references

to other handles. DO was the inspiration for the *ITU-T X.1255 (2013)* framework which introduced an abstract *Digital Entity Interface Protocol* for managing such objects programmatically, first realised by the Digital Object Interface Protocol (DOIP) (*Reilly, 2009*).

In brief, the structure of a FAIR Digital Object (FDO) is to, given a *persistent identifier* (PID) such as a DOI, resolve to a *PID Record* that gives the object a *type* along with a mechanism to retrieve its *bit sequences*, *metadata* and references to further programmatic *operations*. The type of an FDO (itself an FDO) defines attributes to semantically describe and relate such FDOs to other concepts (typically other FDOs referenced by PIDs). The premise of systematically building an ecosystem of such digital objects is to give researchers a way to organise complex digital entities, associated with identifiers, metadata, and supporting automated processing (*Wittenburg et al., 2019*). As mentioned previously, this ecosystem is envisioned to consist of a wide variety of digital entities and contextual information ranging from software to articles to even descriptions of experimental infrastructures (*Azeroual et al., 2022*). Recently, it has been noted that the practical use of FDOs to achieve interoperability requires governance in particular with respect to assessing such interoperability (*Wilkinson et al., 2023*).

FDOs have been recognised by the European Open Science Cloud (EOSC) as a suggested part of its Interoperability Framework (*Corcho et al., 2021*), in particular for deploying active and interoperable FAIR resources that are *machine actionable*. Development of the FDO concept continued within Research Data Alliance (RDA) groups and EU projects like GO-FAIR, concluding with a set of guidelines for implementing FDO (*Bonino et al., 2019*). The FAIR Digital Objects Forum has since taken over the maturing of FDO through focused working groups which have currently drafted several more detailed specification documents (*FAIR Digital Objects, 2022b*).

### FDO approaches

FDO is an evolving concept. A set of FDO Demonstrators (*Wittenburg et al., 2022*) highlights how current adapters are approaching implementations of FDO from different angles:

- Building on the Digital Object concept, using the simplified *DOIPV2.0 (2018)* specification, which detail how to exchange JSON objects through a text-based protocol[1] (usually TCP/IP over TLS). The main DOIP operations are retrieving, creating and updating digital objects. These are mostly realised using the reference implementation Cordra (*Tupelo-Schneck & Lannom, 2022*). FDO types are registered in the local Cordra instance, where they are specified using JSON Schema (*Wright et al., 2022*) and PIDs are assigned using the Handle system. Several type registries have been established.
- Following a Linked Data approach, but using the DOIP protocol, *e.g.*, using JSON-LD and schema.org within DOIP in Materal Sciences archives (*Riccardi et al., 2022*).
- Approaching the FDO principles from existing Linked Data practices on the Web, *e.g.*, WorkflowHub use of RO-Crate and schema.org (*Soiland-Reyes et al., 2022b*).

---

[1] For a brief introduction to DOIP 2.0, see *CNRI (2023a)*

From this it becomes apparent that there is a potentially large overlap between the goals and approaches of FAIR Digital Objects and Linked Data, which we will cover in section *From the Semantic Web to Linked Data*.

### Next steps for FDO

The FAIR Digital Object Forum (*FAIR Digital Objects, 2022a*) working groups have prepared detailed requirement documents (*FAIR Digital Objects, 2022b*) setting out the path for realising FDOs, named *FDO Recommendations*. As of 2023-06-17, most of these documents are open for public review, while some are still in draft stages for internal review. We provide an overview of these documents in 'An Overview of Upcoming FDO Specifications'. These documents clarify the future aims and focus of FAIR Digital Objects (*Lannom et al., 2022b*). Except for the DOIP endorsement, all of these documents are conceptual, in the sense that they permit any technical implementation of FDO, if used according to the recommendations. Going forward, a key strategy of the Forum is the use of profiles to help define specific attributes in metadata that are necessary for domains or application contexts. However, these are not yet fully implemented in the implementations considered here.

Existing FDO implementations (*Wittenburg et al., 2022*) are thus not fully consolidated in choices such as protocols, type systems and serialisations—this divergence and corresponding additional technical requirements mean that FDOs are not yet in a single ecosystem.

### From the Semantic Web to Linked Data

In order to describe *Linked Data* as it is used today, we'll start with an (opinionated) description of the evolution of its foundation, the *Semantic Web*.

### A brief history of the Semantic Web

The **Semantic Web** was developed as a vision by *Berners-Lee & Fischetti (1999)*, at a time that the Web had already become widely established for information exchange, being a global set of hypermedia documents which are cross-related using universal links in the form of URLs. The foundations of the Web (*e.g.*, URLs, HTTP, SSL/TLS, HTML, CSS, ECMAScript/JavaScript, media types) were standardised by W3C, Ecma, IETF and later WHATWG. The goal of Semantic Web was to further develop the machine-readable aspects of the Web, in particular adding *meaning* (or semantics) to not just the link relations, but also to the *resources* that the URLs identified, and for machines thus being able to meaningfully navigate across such resources, *e.g.*, to answer a particular query.

Through W3C, the Semantic Web was realised with the Resource Description Framework (RDF) (*Schreiber & Raimond, 2014*) that used *triples* of subject-predicate-object statements, with its initial serialisation format (*Lassila & Swick, 1999*) being RDF/XML (XML was at the time seen as a natural data-focused evolution from the document-centric SGML and HTML).

While triple-based knowledge representations were not new (*Stanczyk, 1987*), the main innovation of RDF was the use of global identifiers in the form of URIs[2] as the primary identifier of the *subject* (what the statement is about), *predicate* (relation/attribute of the

---

[2] URIs (*Berners-Lee, Fielding & Masinter, 2005*) are generalised forms of URLs that include locator-less identifiers such as ISBN book numbers (URNs). The distinction between locator-full and locator-less identifiers have weakened in recent years (*OCLC, Inc, 2010*), for instance DOI identifiers now are commonly expressed with the prefix https://doi.org/ rather than as URNs with `info:doi:` given that the URL/URN gap has been bridged by HTTP resolvers and the use of Persistent Identifiers (PIDs) (*Juty, Le Novere & Laibe, 2011*). RDF 1.1 formats use Unicode to support *IRIs* (*Dürst Martin & Suignard, 2005*), which extends URIs to include international characters and domain names.

[3] URIs can also identify *non-information resources* for any kind of physical object (*e.g.*, people), such identifiers can resolve with `303 See Other` redirections to a corresponding *information resources* (*Sauermann et al., 2008*).

subject) and *object* (what is pointed to). By using URIs not just for documents[3], the Semantic Web builds a self-described system of types and properties, where the meaning of a relation can be resolved by following its hyperlink to the definition within a *vocabulary*. By applying these principles as well to any kind of resource that could be described at a URL, this then forms a global distributed Semantic Web.

The early days of the Semantic Web saw fairly lightweight approaches with the establishment of vocabularies such as FOAF (to describe people and their affiliations) and Dublin Core (for bibliographic data). Vocabularies themselves were formalised using RDFS or simply as human-readable HTML web pages defining each term. The main approach of this *Web of Data* was that a URI identified a *resource* (*e.g.*, an author) with a HTML *representation* for human readers, along with a RDF representation for machine-readable data of the same resource. By using content negotiation in HTTP, the same identifier could be used in both views, avoiding `index.html` *vs* `index.rdf` exposure in the URLs. The concept of *namespaces* gave a way to give a group of RDF resources with the same URI base from a Semantic Web-aware service a common *prefix*, avoiding repeated long URLs.

The mid-2000s saw large academic interest and growth of the Semantic Web, with the development of more formal representation system for ontologies, such as OWL (*W3C OWL Working Group, 2012*), allowing complex class hierarchies and logic inference rules following *open world* paradigm. More human-readable syntaxes for RDF such as Turtle evolved at this time, and conferences such as ISWC (*Horrocks & Hendler, 2002*) gained traction, with a large interest in knowledge representation and logic systems based on Semantic Web technologies evolving at the same time.

Established Semantic Web services and standards include: SPARQL (*SPARQL Working Group, 2013*) (pattern-based triple queries), named graphs (*Wood, Cyganiak & Lanthaler, 2014*) (triples expanded to *quads* to indicate statement source or represent conflicting views), triple/quad stores (graph databases such as OpenLink Virtuoso, GraphDB, 4Store), mature RDF libraries (including Redland RDF, Apache Jena, Eclipse RDF4J, RDFLib, RDF.rb, rdflib.js), and graph visualisation.

RDF is one way to implement *knowledge graphs*, a system of named edges and nodes[4] (*Nurdiati & Hoede, 2008*), which when used to represent a sufficiently detailed model of the world, can then be queried and processed to answer detailed research questions. The creation of RDF-based knowledge graphs grew particularly in fields like bioinformatics, *e.g.*, for describing genomes and proteins (*Goble & Stevens, 2008*; *Williams et al., 2012*). In theory, the use of RDF by the life sciences would enable interoperability between the many data repositories and support combined views of the many aspects of bio-entities—however in practice most institutions ended up making their own ontologies and identifiers, for what to the untrained eye would mean roughly the same. One can argue that the toll of adding the semantic logic system of rich ontologies meant that small, but fundamental, differences in opinion (*e.g.*, *should a gene identifier signify just the particular DNA sequence letters, or those letters as they appear in a particular position on a human chromosome?*) led to large differences in representational granularity, and thus needed different identifiers.

[4] In RDF, each triple represent an edge that is named using its property URI, and the nodes are subject/object as URIs, blank nodes or (for objects) typed literal values (*Schreiber & Raimond, 2014*).

Facing these challenges, thanks to the use of universal identifiers in the form of URIs, *mappings* could retrospectively be developed not just between resources, but also across vocabularies. Such mappings can be expressed themselves using lightweight and flexible RDF vocabularies such as SKOS (*Isaac & Summers, 2009*) (*e.g.*, `dct:title skos:closeMatch schema:name` to indicate near equivalence of two properties). Exemplifying the need for such cross-references, automated ontology mappings have identified large potential overlaps like 372 definitions of `Person` (*Hu et al., 2011*).

The move towards *Open Science* data sharing practices did from the late 2000s encourage knowledge providers to distribute collections of RDF descriptions as downloadable *datasets*[5], so that their clients can avoid thousands of HTTP requests for individual resources. This enabled local processing, mapping and data integration across datasets (*e.g.*, Open PHACTS (*Groth et al., 2014*)), rather than relying on the providers' RDF and SPARQL endpoints (which could become overloaded when handling many concurrent, complex queries).

With these trends, an emerging problem was that adopters of the Semantic Web primarily utillised it as a set of graph technologies, with little consideration to existing Web resources. This meant that links stayed mainly within a single information system, with little URI reuse even with large term overlaps (*Kamdar, Tudorache & Musen, 2017*). Just like *link rot* affect regular Web pages and their citations from scholarly communication (*Klein et al., 2014*), a majority of described RDF resources in the Linked Open Data (LOD) Cloud's gathering of more than thousand datasets do not actually link to (still) downloadable (*dereferenceable*) Linked Data (*Polleres et al., 2020*). Another challenge facing potential adopters is the plethora of choices, not just to navigate, understand and select to reuse the many possible vocabularies and ontologies (*Carriero et al., 2020*), but also technological choices on RDF serialisation (at least 7 formats), type system (RDFS (*Guha & Brickley, 2014*), OWL (*W3C OWL Working Group, 2012*), OBO (*Tirmizi et al., 2011*), SKOS (*Isaac & Summers, 2009*)), and deployment challenges (*Sauermann et al., 2008*) (*e.g.*, hash *vs* slash in namespaces, HTTP status codes and PID redirection strategies).

### Linked Data: rebuilding the Web of Data

The **Linked Data** (LD) concept (*Bizer, Heath & Berners-Lee, 2009*) was kickstarted as a set of best practices (*Berners-Lee, 2006*) to bring the Web aspect of the Semantic Web back into focus. Crucial to Linked Data is the *reuse of existing URIs*, rather than making new identifiers. This means a loosening of the semantic restrictions previously applied, and an emphasis on building navigable data resources, rather than elaborate graph representations.

Vocabularies like schema.org evolved not long after, intended for lightweight semantic markup of existing Web pages, primarily to improve search engines' understanding of types and embedded data. In addition to several such embedded *microformats* (*OGP, 2022*; *Sporny et al., 2015*; *Microdata, 2023*), we find JSON-LD (*Sporny et al., 2020*) as a Web-focused RDF serialisation that aims for improved programmatic generation and consumption, including from Web applications. JSON-LD is as of 2023-05-18 used[6] by 45% of the top 10 million websites (*W3Techs, 2023*).

[5] *Datasets* that distribute RDF graphs should not be confused with RDF Datasets used for partitioning *named graphs*.

[6] Presumably this large uptake of JSON-LD is mainly for the purpose of Search Engine Optimisation (SEO), with typically small amounts of metadata which may not constitute Linked Data as introduced above, however this deployment nevertheless constitute machine-actionable structured data.

Recently there has been a renewed emphasis to improve the *Developer Experience* (*Verborgh, 2018*) for consumption of Linked Data, for instance RDF Shapes—expressed in SHACL (*Kontokostas & Knublauch, 2017*) or ShEx (*Baker & Hommeaux, 2019*)—can be used to validate RDF Data (*Gayo et al., 2017*; *Thornton et al., 2019*) before consuming it programmatically, or reshaping data to fit other models. While a varied set of tools for Linked Data consumptions have been identified, most of them still require developers to gain significant knowledge of the underlying Semantic Web technologies, which hampers adaption by non-LD experts (*Klímek et al., 2019*), which then tend to prefer non-semantic two-dimensional formats such as CSV files.

A valid concern is that the Semantic Web research community has still not fully embraced the Web, and that the "final 20%" engineering effort is frequently overlooked in favour of chasing new trends such as Big Data and AI, rather than making powerful Linked Data technologies available to the wider groups of Web developers (*Verborgh & Vander Sande, 2020*). One bridging gap here by the Linked Data movement has been "Linked Data by stealth" approaches such as structured data entry spreadsheets powered by ontologies (*Wolstencroft et al., 2011*), the use of Linked Data as part of REST Web APIs (*Page, De Roure & Martinez, 2011*), and as shown by the big uptake by publishers to annotate the Web using schema.org (*Bernstein, Hendler & Noy, 2016*), with vocabulary use patterns documented by copy-pastable JSON-LD examples, rather than by formalised ontologies or developer requirements to understand the full Semantic Web stack.

Linked Data provides technologies that have evolved over time to satisfy its primary purpose of data interoperability. The needs to embrace the Web and developer experience have been central lessons learned. In contrast, FDO is a new approach with many different potential paths forward, and having a partial overlap with the aims of Linked Data.

## METHOD

Our main motivation for this article is to investigate how FAIR Digital Objects may differ from the learnt experiences of Linked Data and the Web. We also aim to reflect back from FDO's motivation of machine-actionability to consider the Web as a distributed computational system.

To better understand the relationship between the FDO framework and other existing approaches, we use the following for analysis:

1) An Interoperability Framework and Distributed Platform for Fast Data Applications (*Delgado, 2016*), which proposes quality measurements for comparing how frameworks support interoperability, particularly from a service architectural view.

2) The FAIR Digital Object guidelines (*Bonino et al., 2019*), validated against its current implementations for completeness.

3) A Comparison Framework for Middleware Infrastructures (*Zarras, 2004*), which suggest dimensions like openness, performance and transparency, mainly focused on remote computational methods.

4) Cross-checks against RDA's FAIR Data Maturity Model (*Bahim et al., 2020*) to find how the FAIR principles are achieved in FDO, in particular considering access, sharing and openness.

5) EOSC Interoperability Framework (*Corcho et al., 2021*) which gives recommendations for technical, semantic, organisational and legal interoperability, particularly from a metadata perspective.

Conceptual frameworks 1, 3, 5 consider more general views of interoperability between systems, whereas frameworks 2 and 4 are developed specifically for addressing FAIR principles.

The reason for this wide-ranged comparison is to exercise the different dimensions that together form FAIR Digital Objects: Data, Metadata, Service, Access, Operations, Computation. We have left out further considerations on type systems, persistent identifiers and social aspects as principles and practices within these dimensions are still taking form within the FDO community.

Some of these frameworks invite a comparison on a conceptual level, while others relate better to implementations and current practices. For conceptual comparisons we consider FAIR Digital Objects and the Web broadly. For implementations, we contrast the main FDO realisation using the DOIPv2 protocol (*DOIPV2.0, 2018*) against Linked Data as implemented in general practice[7].

For all our comparisons, our process was to perform a mapping between the relevant specifications and/or implementation and the given conceptual model through detailed reading of the defining documents. We aim in all cases for traceability between the given specification and our mapping such that readers can validate our analysis.

## RESULTS

### Considering FDO/Web as interoperability framework for Fast Data

The Interoperability Framework for Fast Data Applications (*Delgado, 2016*) categorises interoperability between applications along six strands, covering different architectural levels: from *symbiotic* (agreement to cooperate) and *pragmatic* (ability to choreograph processes), through *semantic* (common understanding) and *syntactic* (common message formats), to low-level *connective* (transport-level) and *environmental* (deployment practices).

We have chosen to investigate using this framework as it covers the higher levels of the OSI Model (*Stallings, 1990*) better with regards to automated machine-to-machine interaction (and thus interoperability), which is a crucial aspect of the FAIR principles. In Table 1 we use the interoperability framework to compare the current FAIR Digital Object approach against the Web and its Linked Data practices.

### Observations

Based on the analysis shown in Table 1, we draw the following conclusions:

The Web has already showed us how one can compose workflows of hetereogeneous Web Services (*Wolstencroft et al., 2013*). However, this is mostly done *via* developer or

---

[7] For further background on FDO implemented with Linked Data see (*Bonino da Silva Santos, Guizzardi & Sales, 2022*; *Soiland-Reyes et al., 2022a*).

**Table 1 Considering FDO and Web according to the quality levels of the Interoperability Framework for Fast Data (*Delgado, 2016*).**

| Quality | FDO w/DOIP | Web w/Linked Data |
|---|---|---|
| **Symbiotic:** *to what extent multiple applications can agree to interact, align, collaborate or cooperate* | The purpose of FDO is to enable federated machine actionable digital objects for scholarly purposes, in practice this also requires agreement of compatibility between FDO types. FDO encourages research communities to develop common type registries to be shared across instances. In current DOIP practice, each service have their own types, attributes and operations. The wider symbiosis is consistent use of PIDs. | The Web is loosely coupled and encourages collaboration and linking by URL. In practice, REST APIs (*Fielding, 2000*) end up being mandated centrally by dominant (often commercial) providers (*Fielding et al., 2017*), and the clients are required to use each API as-is with special code per service. Use of Linked Data enables common tooling and semantic mapping across differences. |
| **Pragmatic:** *using interaction contracts so processes can be choreographed in workflows* | FDO types and operations enable detailed choreography (Canonical Workflows; *CWFR Group (2021)*). `0.TYPE/DOIPOperation` has lightweight definition of operation, `0.DOIP/Request` or `0.DOIP/Response` may give FDO Type or any other kind of "specifics" (incl. human readable docs). Semantics/purpose of operations not formalised (similar operations can be grouped with `0.DOIP/OperationReference`). | "Follow your nose" crawler navigation, which may lead to frequent dead ends. Operational composition, typically within a single API provider, documented by OpenAPI 3 (*Miller et al., 2021*), schema.org Actions (*Schema.org, 2022b*), WSDL/SOAP (*Liu & Booth, 2007*), but frequently just as human-readable developer documentation with examples. |
| **Semantic:** *ensuring consistent understanding of messages, interoperability of rules, knowledge and ontologies* | FDO semantic enable navigation and typing. Every FDO has a type. Types maintained in FDO Type registries, which may add additional semantics, *e.g.*, the ePIC PID-InfoType for Model. No single type semantic, Type FDOs can link to existing vocabularies & ontologies. JSON-LD used within some FDO objects (*e.g.*, DISSCO Digital Specimen, NIST Material Science schema) (*Wittenburg et al., 2022*) | Lightweight HTTP semantics for authenticity/navigation. Semantic Type not commonly expressed on PID/header level, may be declared within Linked Data metadata. Semantic of type implied by Linked Data formats (*e.g.*, OWL2, RDFS), although choice of type system may not be explicit. |
| **Syntactic:** *serialising messages for digital exchange, structure representation* | DOIP serialise FDOs as JSON, metadata commonly use JSON, typed with JSON Schema. Multiple byte stream attachments of any media type. | Textual HTTP headers (including any signposting), single byte stream of any media type, *e.g.*, Linked Data formats (JSON-LD, Turtle, RDF/XML) or embedded in document (HTML with RDFa, JSON-LD or Microdata). XML was previously the main syntax used by APIs, JSON is now dominant. |
| **Connective:** *transferring messages to another application, e.g., wrapping in other protocols* | *DOIPV2.0 (2018)* is transport-independent, commonly TLS TCP/IP port 9000, DOIP over HTTP (*CNRI, 2023b*) | HTTP/1.1, TCP/IP port 80 (*Fielding et al., 1999*); HTTP/1.1+TLS, TCP/IP 443 (*Rescorla, 2000*); HTTP/2, as HTTP/1* but binary (*Belshe, Peon & Thomson, 2015*); HTTP/3, like HTTP/2+TLS but UDP (*Bishop, 2022*) |
| **Environmental:** *how applications are deployed and affected by its environment, portability* | Main DOIP implementation is Cordra, which can be single-instance or distributed. Cordra storage backends include file system, S3, MongoDB (itself scalable). Unique DOIP protocol can be hard to add to existing Web application frameworks, although proxy services have been developed (*e.g.*, B2SHARE adapter). | HTTP services widely deployed in a myriad of ways, ranging from single instance servers, horizontally & vertically scaled application servers, to multi-cloud Content-Delivery Networks (CDN). Current scalable cloud technologies for Web hosting may not support HTTP features previously seen as important for Semantic Web, *e.g.*, content negotiation and semantic HTTP status codes. |

human interaction (*Lamprecht et al., 2021*). Similarly, FDO does not enable automatic composition because operation semantics are not well defined. There is a question as to whether the extensive documentation and broad developer usage that is available for Web APIs could potentially be utilised for FDO.

**Table 2** Mapping the Metamodel concepts from the Interoperability Framework for Fast Data (*Delgado, 2016*) to equivalent concepts for FDO and Web.

| Metamodel concept | FDO/DOIP concept | Web/HTTP concept |
| --- | --- | --- |
| Resource | FDO/DO | Resource |
| Service | DOIP service | Server/endpoint |
| Transaction | (not supported) | Conditional requests, 409 `Conflict` |
| Process | Extended operations | (primarily stateless), 100 `Continue`, 202 `Accepted` |
| Operation | DOIP Operation | Method, query parameters |
| Request | DOIP Request | Request |
| Response | DOIP Response | Response |
| Message | Segment, `requestId` | Message, Representation |
| Channel | DOIP Transport protocol (*e.g.*, TCP/IP, TLS). JSWS signatures. | TCP/IP, TLS, UDP |
| Protocol | DOIP 2.0, ++ | HTTP/1.1, HTTP/2, HTTP/3 |
| Link | PID/Handle | URL |

A difference between Web technologies and FDO is the stringency of the requirements for both syntax and semantics. Whereas the Web allows many different syntactic formats (*e.g.*, from HTML to XML, PDFs), FDO realised with DOIP requires JSON. On the semantic front, FDO mandates that every object have a well-defined type and structured form. This is clearly not the case on the Web.

In terms of connectivity and the deployment of applications, the Web has a plethora of software, services, and protocols that are widely deployed. These have shown interoperability. The Web standards bodies (*e.g.*, IETF and W3C) follow the OpenStand principles (*OpenStand, 2017*) to embrace openness, transparency, and broad consensus. In contrast, FDO has a small number of implementations and corresponding protocols, although with a growing community, as evidenced at the first international FDO conference (*Loo, 2022*). This is not to say that it is not worth developing further Handle +DOIP implementations in the future, but we note that the current FDO functionality can easily be implemented using Web technologies, even as DOIP-over-HTTP (*CNRI, 2023b*).

It is also a question as to whether a highly constrained protocol revolving around persistent identifiers is in fact necessary. For example, DOIs are mostly resolved on the web using HTTP redirects with the common https://doi.org/ prefix, hiding their Handle nature as an implementation detail (*DOI, 2019*).

### Mapping of metamodel concepts

The Interoperability Framework for Fast Data also provides a brief *metamodel* which we use in Table 2 to map and examplify corresponding concepts in FDO's DOIP realization and the Web using HTTP semantics (*Fielding, Nottingham & Reschke, 2022*).

From this mapping[8] we can identify the conceptual similarities between DOIP and HTTP, often with common terminology. Notable are that neither DOIP or HTTP have strong support for transactions (explored further in section *Comparing FDO and Web as middleware infrastructures*), as well that HTTP has poor direct support for processes, as the Web is primarily stateless by design.

[8] A SKOS mapping (*Isaac & Summers, 2009*) is provided as part of the RO-Crate for this article (*Soiland-Reyes, 2023a*).

## Assessing FDO implementations

The FAIR Digital Object guidelines (*Bonino et al., 2019*) sets out recommendations for FDO implementations. Note that the proposed update to FDO specification (*Anders et al., 2023b*) clarifies these definitions with equivalent identifiers[9] and relates them to further FDO requirements such as FDO Data Type Registries.

In Table 3, we evaluate completeness of the guidelines in two current FDO realisations: (1) DOIPv2 (*DOIPV2.0, 2018*) and (2) Linked Data Platform (*Speicher, Arwe & Malhotra, 2015*), as proposed by *Bonino da Silva Santos, Guizzardi & Sales (2022)*. We provide our analysis of each realisation with respect to the FDO Guideline and also provide suggestions for that realisation to meet the given guideline.

A key insight from this is that simply using DOIP does not achieve many of the FDO guidelines. Rather the guidelines set out how a protocol like DOIP should be used to achieve FAIR Digital Object goals. The DOIP Endorsement (*Schwardmann et al., 2022*) set out that to comply, DOIP must be used according to the set of FDO requirement documents and notes *Achieving FDO compliance requires more than DOIP and full compliance is thus left to system designers*. Likewise, a Linked Data approach will need to follow the same FDO requirements to actually comply as an FDO implementation.

### *Observations*

- G1 and G2 call for stability and trustworthiness. While the foundations of both DOIP and Linked Data approaches are now well established—the FDO requirements and in particular how they can be implemented are still taking shape and subject to change.

- Machine actionability (G4, G6) is a core feature of both FDOs and Linked Data. Conceptually they differ in the way types and operations are discovered, with FDO seemingly more rigorous. In practice, however, we see that DOIP also relies on dynamic discovery of operations and that operation expectations for types (FDOF7) have not yet been defined.

- FDO proposes that types can have additional operations beyond CRUD (FDOF5, FDOF6), while Linked Data mainly achieves this with RESTful patterns using CRUD on additional resources, *e.g.*, `order/152/items`. These are mainly stylistics but affect the architectural view—FDOs have more of an object-oriented approach.

- FDO puts strong emphasis on the use of PIDs (FDOF1, FDOF2, FDOF3, FDOF5), but in current practice DOIP use local types, local extended operations (FDOF5) and attributes (FDOF4) that are not bound to any global namespace.

- Linked Data have a strong emphasis on semantics (FDOF8), and metadata schemas developed by community agreements (FDOF10). FDO types need to be made reusable across servers.

- While FDO recommends nested metadata FDOs (FDOF8, FDOF9), in practice this is not found (or linked with custom keys), particularly due to lack of namespaces and the favouring of local types rather than type/property re-use. Linked Data frequently have multiple representations, but often not sufficiently linked (link relation `alternate`

[9] Newer (*Anders et al., 2023b*) renames `FDOF*` to `FDO-GR*` but follows same ordering.

**Table 3** Checking FDO guidelines (*Bonino et al., 2019*; *Anders et al., 2023b*) against its current implementations as DOIP (*DOIPV2.0, 2018*) and linked data platform (LDP) (*Bonino da Silva Santos, Guizzardi & Sales, 2022*), with suggestions for required additions.

| FDO guideline | DOIP 2.0 | FDO suggestions | Linked data platform | LDP suggestion |
|---|---|---|---|---|
| G1: *invest for many decades* | Handle system stable for 20 years, DOIP 2.0 since 2017. | Ensure FDO types will not be protocol-bound as DOIP might be updated/replaced | HTTP stable for 30 years, Semantic Web for 20 years. http://URIs mostly replaced by https://. | Keep flexibility of RDF serialisation formats which may change more frequently |
| G2: *trustworthiness* | DOI/Handle trusted by all major academic publishers and data repositories. DOIP relatively unknown, in effect only one implementation. | Further promote DOIP and justify its benefits. Build tutorials and OSI open source implementations. Standardise DOIP-over-HTTP alternative. | JSON-LD used by half of all websites (*W3Techs, 2023*), however previous bad experiences with Semantic Web may deter adopters | Ensure simplicity for end developers, rather than semantic overengineering. Example-driven documentation. |
| G3: *follows FAIR principles* | See Table 5 | Ensure all FAIR principles are covered, build complete examples. | Touched briefly, see Table 5 | Add explicit expression to show each FAIR principle fulfilled. |
| G4: *machine actionability* | CRUD and extension operations dynamically listed (see Table 4) | Specify which operations should work for a given type, to reduce need for dynamic lookup. Specify input/output expectations formally (*e.g.*, JSON Schema). | HTTP CRUD operations, Open API (see Table 4) | Document operations so client can make subsequent HTTP calls. |
| G5: *abstraction principle* | Handle PIDs as abstraction base. DOIP operations can use any transport protocol. | Document transport protocols as FDOs, recommend which transport to use. | URI as abstraction base. Does not specify PID requirements. | Give stronger deployment recommendations. |
| G6: *stable binding between entities* | Machine-navigation through PIDs and operations specified per type. Unclear when metadata field is a PID or plain text. | Make datatype of fields explicit to support navigation. | Machine-navigation through URIs *via* properties and types. Unclear when URI should be followed or is just identifier, but always distinct from plain text. | |
| G7: *encapsulation* | Operations discovered at runtime (`0.DOIP/Op.ListOperations`). | Allow method discovery by type FDOs in advance (*Lannom et al., 2022c*). | HTTP methods discovered at runtime (`OPTIONS`), indempotent methods attempted directly. Unsupported methods reported using LDP constraints to human-readable text. | Declare supported methods in advance, *e.g.*, OpenAPI (*Miller et al., 2021*) |
| G8: *technology independence* | In theory independent, in reality depends on single implementations of Handle system and DOIP | Encourage open source DOIP testbeds and lighter reference implementations | Multiple HTTP implementations, multiple LDP implementations. No FDOF implementations. | Develop demonstrator of FDOF usage based on existing LDP server. |
| G9: *standard compliance* | Handle (*Sun, Lannom & Boesch, 2003*), DOIP (*DOIPV2.0, 2018*). FDO requirements not standardised yet. | Formalise standard process of FDO requirements (*Weiland et al., 2022b*) | HTTP, LDP. However FDOF is not yet standardised. | Formalise FDOF from FDOF-SEM working group. |

| FDO guideline | DOIP 2.0 | FDO suggestions | Linked data platform | LDP suggestion |
|---|---|---|---|---|
| FDOF1: *PID as basis* | Extensive use of Handle system. | Clarify how local testing handles can be used during development, how "temporary" FDOs should evolve (*Anders et al., 2022*). Register `0.DOIP/*` and `0.FDO/*` as actual PIDs. | HTTP URLs as basis for identifiers, but they are frequently not persistent. | Add strong guidance for PID services like w3id and persistence policies (*McMurry et al., 2017*). |
| FDOF2: *PID record w/ type* | Unspecified how to resolve from Handle to DOIP Service (!), in practice `10320/loc`, `0.TYPE/DOIPService`, URL, URL_REPLICA | Document requirements for PID Record | w3id/purl PIDs redirect without giving any metadata. Datacite DOIs content-negotiate to give registered metadata. | Add FAIR Signposting (*Van de Sompel et al., 2022*) at PID provider for minimal PID record |
| FDOF3: *PID resolvable to bytestream & metadata* | Byte stream resolvable (`0.DOIP/Retrieve`), `includeElementData` option can retrieve bytestream or full object structure. No method/attribute defined for separate metadata, only directly in PID Record. Unclear meaning of multiple items and bytestream chunks. | Clarify expectations for multiple items. Recommend chunks to not be used. | URIs resolvable by default. Multiple ways to resolve metadata, unclear preference. | Add FAIR Signposting and preference order. |
| FDOF4: *Additional attributes* | Freetext attribute keys. Attributes should be defined for FDO type. | Require that attribute keys should be PIDs (or have predefined mapping to PIDs). Explicitly allow attributes not already defined in type. | All attributes individually identified. Any Linked Data attributes can be used by URI or with mapped prefix. | Clarify type expectations of required/recommended/optional attributes. |
| FDOF5: *Interface: operation by PID* | Extended operations use PID, but "pid-like" DOIP operations/types are not registered as handles. | Register `0.DOIP/*` and `0.FDO/*` as PIDs. Clarify that operations can be mapped to protocol directly. | CRUD operations used directly in HTTP (*e.g.*, PUT). Unclear how to provide PID for additional operations. | Specify how additional operations should be called over HTTP. |
| FDOF6: *CRUD operations + extensions* | `0.DOIP/Op.Create`, `Op.Retrieve`, `Op.Update`, `Op.Delete` but also `0.DOIP/Op.Search`. | Document | PUT, GET, POST, DELETE, PATCH, HEAD–extension operations (*e.g.*, WebDAV COPY) not used, resource patterns (*Ekuan et al., 2023*) are used instead. | Document how operation resources can be discovered from an LPD container. Document search API. |
| FDOF7: *FDOF Types related to operations* | Not yet formalised, by DOIP discoverable on a given FDO rather than type. PR-TypingFDOs leaves this open. | Add explicit relation between type and operations | OPTIONS per LDP Resource, but not by type. Common types (`ldp:Resource`, `ldp:Container`) indicate LDP support, but are not required. | Always make LDP types explicit in FDO profile. |

(Continued)

| FDO guideline | DOIP 2.0 | FDO suggestions | Linked data platform | LDP suggestion |
|---|---|---|---|---|
| FDOF8: *Metadata as FDO, semantic assertions* | DOIP includes all metadata in PID Record. Separate Metadata FDO need custom property. | Specify a `0.FDO/metadata` or similar to point to Metadata FDOs. | Assertions are always with semantics, using RDF vocabularies. Unspecified how to find additional metadata resources, `rdfs:seeAlso` is common. | Use FAIR Signposting `describedby` link relation to additional metadata PIDs |
| FDOF9: *Different metadata levels* | Defines open-ended "Response Attributes" without namespaces, but mandated as "None" for all CRUD operations. Metadata would need to be bundled within custom FDO types or attributes. Unclear how levels are separated within single FDO representation (may need FDOF8). | Declare which metadata are expected within response attribute or within FDO object. Require PIDs for custom attributes. Define how alternate metadata levels can be represented separately. | Undefined how to handle multiple metadata granularities or domains, alternative LDP containers can present different views on same stored objects. | Define how to navigate to alternate views and their semantic implications, *e.g.* `owl:sameAs` |
| FDOF10: *Metadata schemas by community* | Metadata schemas are in practice managed on single Cordra server as local types, using JSON Schema. | Require types to be FDOs with registered PIDs, implement shared types. | Plethora of existing RDF vocabularies/ontologies managed by larger communities, *e.g.*, OBO Foundry (*Smith et al., 2007*) | Rather document better how individual ad-hoc schemas can be started for prototypes. |
| FDOF11: *FDO collections w/semantic relations* | Collection type undefined by DOIP. Informal use of `HAS_PARTS` Handle attribute (*e.g.*, *Semmler et al., 2022*). | | LDP Containers required by specification, also user-created (eg. `BasicContainer`). | Clarify relation to other collections like DCAT 3 (*Dataset Exchange Working Group, 2023*), Schema.org Dataset, OAI-ORE (*Lagoze et al., 2008*) |
| FDOF12: *Deleted FDO preserve PID w/ tombstone* | Tombstone for deleted resource undefined by DOIP. `0.DOIP/Status.104` status code does not distinguish "Not Found" or "Gone" | Formalise tombstone requirements with new FDO type | `410 Gone` recommended, but `404 Not Found` common. No requirement for tombstone serialisation | Formalise tombstone requirements and serialisation |

(*Nottingham, 2017*)) or related (`prov:specializationOf` from *Lebo, McGuinness & Sahoo (2013)*).

- FDO collections are not yet defined for DOIP, while Linked Data seemingly have too many alternatives. LDP has specific native support for containers.
- Tombstones for deleted resources are not well supported, nor specified, for either approach, although the continued availability of metadata when data is removed is a requirement for FAIR principles (RDA-A2-01M).
- DOIP supports multiple chunks of data for an object (FDOF3), while Linked Data can support content-negotiation. In either case it can be unclear to clients what is the meaning or equivalence of any additional chunks.

## Comparing FDO and Web as middleware infrastructures

In this section, we take the perspective that FDO principles are in effect proposing a global infrastructure of machine-actionable digital objects. As such we can consider implementations of FDO as **middleware infrastructures** for programmatic usage, and can evaluate them based on expectations for client and server developers.

We argue that the Web, with its now ubiquitous use of REST API (*Fielding, 2000*), can be compared as a similar global middleware. Note that while early moves for developing Semantic Web Services (*Fensel et al., 2011*) attempted to merge the Web Service and RDF aspects, we are here considering mainly the current programmatic Web and its mostly light-weight use of three out of possible *five stars Linked Data* (*Hausenblas & Kim, 2012*).

For this purpose, we here utillise the Comparison Framework for Middleware Infrastructures (*Zarras, 2004*) that formalise multiple dimensions of openness, scalability, transparency, as well as characteristics known from Object-oriented programming such as modularity, encapsulation and inheritance.

### *Observations*

Based on the analysis in Table 4, we make the following observations:

- With respect to the aspect of *Performance*, it is interesting to note that the first version of DOIP (*Reilly, 2009*) supported multiplexed channels similar to HTTP/2 (allowing concurrent transfer of several digital objects). Multiplexing was removed for the much simplified DOIP 2.0 (*DOIPV2.0, 2018*). Unlike DOIP 1.0, DOIP 2.0 will require a DO response to be sent back completely, as a series of segments (which again can be split the bytes of each binary *element* into sized *chunks*), before transmission of another DO response can start on the transport channel. It is unclear what is the purpose of splitting a binary into chunks on a channel which no longer can be multiplexed and the only property of a chunk is its size[10].

- HTTP has strong support for scalability and caching, but this mostly assumes read-operations from static resources. FDO has no view on immutability or validity of retrieved objects, but this should be taken into consideration to support large-scale usage.

- HTTP optimisations for performance (*e.g.*, HTTP/2, multiplexing) is largely used for commercial media distribution (*e.g.*, Netflix), and not commonly used by providers of FAIR data

- Cloud deployment of Web applications give many middleware benefits (Scalability, Distribution, Access transparency, Location transparancy)—it is unclear how DOIP as a custom protocol would perform in a cloud setting as most of this infrastructure assumes HTTP as the protocol.

- Programmatically the Web is rather unstructured as middleware, as there are many implementation choices. Usually it is undeclared what to expect for a given URI/service, and programmers follow documented examples for a particular service rather than automated programmatic exploration across providers. This mean one can consider the Web as an ecosystem of smaller middlewares with commonalities.

[10] Although it is possible with `O.DOIP/Op.Retrieve` to request only particular individual elements of an DO (*e.g.*, one file), unlike HTTP's `Range` request, it is not possible to select individual chunks of an element's bytestream.

**Table 4 Comparing FAIR Digital Object (with the DOIP 2.0 protocol (*DOIPV2.0, 2018*)) and Web technologies (using Linked Data) as middleware infrastructures (*Zarras, 2004*).**

| Quality | FDO w/DOIP | Web w/Linked Data |
|---|---|---|
| **Openness:** *framework enable extension of applications* | FDOs can be cross-linked using PIDs, pointing to multiple FDO endpoints. Custom DOIP operations can be exposed, although it is unclear if these can be outside the FDO server. PID minting requires Handle.net prefix subscription, or use of services like Datacite, B2Handle. | The Web is inherently open and made by cross-linked URLs. Participation requires DNS domain purchase (many free alternatives also exists). PID minting can be free using PURL/ARK services, or can use DOI/Handle with HTTP redirects. |
| **Scalability:** *application should be effective at many different scales* | No defined methods for caching or mirroring, although this could be handled by backend, depending on exposed FDO operations (*e.g.*, Cordra can scale to multiple backend nodes) | Cache control headers reduce repeated transfer and assist explicit and transparent proxies for speed-up. HTTP GET can be scaled to world-population-wide with Content-Delivery Networks (CDNs), while write-access scalability is typically manage by backend. |
| **Performance:** *efficient and predictable execution* | DOIP has been shown moderately scalable to 100 millions of objects, create operation at 900 requests/second. DOIP protocol is reusable for many operations, multiple requests may be answered out of order (by requestId). Multiple connections possible. Setup is typically through TCP and TLS which adds latency. | HTTP traffic is about 10% of global Internet traffic, excluding video and social networks (*Sandvine, 2022*). HTTP 1 connections are serial and reusable, and concurrent connections is common. HTTP/2 adds asynchronous responses and multiplexed streams (*Belshe, Peon & Thomson, 2015*) but still has TCP+TLS startup costs. For reduced latency, HTTP/3 (*Bishop, 2022*) use QUIC (*Iyengar & Thomson, 2021*) rather than TCP, already adapted heavily (30% of EMEA traffic) of which Instagram & Facebook video is the majority of traffic (*Joras & Chi, 2020*). |
| **Distribution transparency:;;** *application perceived as a consistent whole rather than independent elements.* | Each FDO is accessed separately along with its components (typically from the same endpoint). FDOs should provide the mandatory kernel metadata fields. FDOs of the same declared type typically share additional attributes (although that schema may not be declared). DOIP does not enforce metadata typing constraints, this need to be established as FDO conventions. | Each URL accessed separately. Common HTTP headers provide basic metadata, although it is often not reliable. A multitude of schemas and serializations for metadata exists, conventions might be implied by a declared profile or certain media types. Metadata is not always machine findable, may need pre-agreed API URI Templates (*Gregorio et al., 2012*), content-negotiation (*MDN, 2023*) or FAIR Signposting (*Van de Sompel et al., 2022*). |
| **Access transparency:** *local/remote elements accessed similarly* | FDOs should be accessed through PID indirection, this means difficult to make private test setup. Commonly a fixed DOIP server is used directly, which permits local non-PID identifiers. | Global HTTP protocol frequently used locally and behind firewalls, but at risk of non-global URIs (*e.g.*, http://localhost/object/1) and SSL issues (*e.g.*, self-signed certificates, local CAs) |
| **Location transparency:** *elements accessed without knowledge of physical location* | FDOs always accessed through PIDs. Multiple locations possible in Handle system, can expose geo-info. | PIDs and URL redirects. DNS aliases and IP routing can hide location. Geo-localised servers common for large cloud deployments. |
| **Concurrency transparency:** *concurrent processing without interference* | No explicit concurrency measures. FDO kernel metadata can include checksum and date. | HTTP operations are classified as being stateless/idempotent or not (*e.g.*, PUT changes state, but can be repeated on failure), although these constraints are occasionally violated by Web applications. Cache control, ETag (*e.g.*, checksum) and modification date in HTTP headers allows detection of concurrent changes on a single resource. |
| **Failure transparency:** *service provisioning resilient to failures* | DOIP status codes, *e.g.*, 0.DOIP/Status.104, additional codes can be added as custom attributes | HTTP status codes *e.g.*, 404 Not Found, specific meaning of standard codes can be documented in Open API. Custom codes uncommon. |

| Quality | FDO w/DOIP | Web w/Linked Data |
|---|---|---|
| **Migration transparency:** *allow relocating elements without interfering application* | Update of PID record URLs, indirection through `0.TYPE/DOIPServiceInfo` (not always used consistently). No redirection from DOIP service. | HTTP `30x` status codes provide temporary or permanent redirections, commonly used for PURLs but also by endpoints. |
| **Persistence transparency:** *conceal deactivation/reactivation of elements from their users* | FDO requires use of PIDs for object persistence, including a tombstone response for deleted objects. There is no guarantee that an FDO is immutable or will even stay the same type (note: CORDRA extends DOIP with version tracking). | URLs are not required to persist, although encouraged (*Berners-Lee, 1998*). Persistence requires convention to use PIDs/PURLs and HTTP `410 Gone`. An URL may change its content, change in type may sometimes force new URLs if exposing extensions like `.json`. Memento (*Van de Sompel, Nelson & Sanderson, 2013*) expose versioned snapshots. WebDAV `VERSION-CONTROL` method (*Clemm et al., 2002*) (used by SVN). |
| **Transaction transparency:** *coordinate execution of atomic/isolated transactions* | No transaction capabilities declared by FDO or DOIP. Internal synchronisation possible in backend for Extended operations. | Limited transaction capabilities (*e.g.*, `If-Unmodified-Since`) on same resource. WebDAV locking mechanisms (*Dusseault, 2007*) with `LOCK` and `UNLOCK` methods. |
| **Modularity:** *application as collection of connected/distributed elements* | FDOs are inherently modular using global PID spaces and their cross-references. In practice, FDOs of a given type are exposed through a single server shared within a particular community/institution. | The Web is inherently modular in that distributed objects are cross-referenced within a global URI space. In practice, an API's set of resources will be exposed through a single HTTP service, but modularity enables fine-grained scalability in backend. |
| **Encapsulation:** *separate interface from implementation. Specify interface as contract, multiple implementations possible* | Indirection by PID gives separation. FDO principles are protocol independent, although it may be unclear which protocol to use for which FDO (although `0.DOIP/Transport` can be specified after already contacting DOIP). Cordra supports native DOIP, DOIP over HTTP and Cordra REST API) | HTTP/1.1 semantics can seamlessly upgrade to HTTP/2 and HTTP/3. `http` *vs* `https` URIs exposes encryption detail*. Implementation details may leak into URIs (*e.g.*, `search.aspx`), countered by deliberate design of URI patterns (*Berners-Lee, 1998*) and PIDs *via* Persistent URLs (PURL). |
| **Inheritance:** *Deriving specialised interface from another type* | DOIP types nested with parents, implying shared FDO structures (unclear if operations are inherited). FDO establishes need for multiple Data Type Registries (*e.g.*, managed by a community for a particular domain). Semantics of type system currently undefined for FDO and DOIP, syntactic types can also piggyback of FDO type's schema (*e.g.*, CORDRA `$ref` use of JSON Schema references (*Wright et al., 2022*)) | Syntactically Media Type with multiple suffixes (*Sporny & Guy, 2023*) (mainly used with `+json`), declaration of subtypes as profiles (RFC6906) *The 'profile' Link Relation Type* (*Wilde, 2013*). In metadata, semantic type systems (RDFS (*Guha & Brickley, 2014*), OWL2 (*W3C OWL Working Group, 2012*), SKOS (*Isaac & Summers, 2009*). OpenAPI 3 (*Miller et al., 2021*) inheritance and Polymorphism. XML `xsd:schemaLocation` or `xsd:type` (*Thompson et al., 2012*), JSON `$schema` (*Wright et al., 2022*), JSON-LD `@context` (*Sporny et al., 2020*). Large number of domain-specific and general ontologies define semantic types, but finding and selecting remains a challenge. |
| **Signal interfaces:** *asynchronous handling of messages* | DOIP 2.0 is synchronous, in FDO async operations undefined. Could be handled as custom jobs/futures FDOs | HTTP/2 multiplexed streams (*Belshe, Peon & Thomson, 2015*), Web Sockets (*Rice et al., 2022*), Linked Data Notifications (*Capadisli & Guy, 2017*), AtomPub (*Gregorio & de Hóra, 2007*), SWORD (*Jones & Jefferies, 2021*), Micropub (*Parecki, 2017*), more typically ad-hoc jobs/futures REST resources |

(Continued)

| Table 4 (continued) | | |
|---|---|---|
| **Quality** | **FDO w/DOIP** | **Web w/Linked Data** |
| **Operation interfaces:** *defining operations possible on an instance, interface of request/response messages* | CRUD predefined in DOIP, custom operations through 0.DOIP/Op.ListOperations (can be FDOs of type 0.TYPE/DOIPOperation, more typically local identifiers like "getProvenance") | CRUD predefined in HTTP methods (*Fielding & Reschke, 2014b*), (extended by registration), URI Templates (*Gregorio et al., 2012*), OpenAPI operations (*Miller et al., 2021*), HATEOAS** incl. Hydra (*Lanthaler, 2021*), schema.org Actions (*Schema.org, 2022b*), JSON HAL (*Kelly, 2016*) & Link headers (RFC8288) (*Nottingham, 2017*) |
| **Stream interfaces:** *operations that can handle continuous information streams* | Undefined in FDO. DOIP can support multiple byte stream elements (need custom FDO type to determine stream semantics) | HTTP 1.1 (*Fielding & Reschke, 2014a*) chunked transfer, HLS (RFC8216) (*Pantos & May, 2017*), MPEG-DASH (*ISO/IEC23009-1, 2022*) |

**Note:**
* The HTTP protocol (port 80) can in theory also upgrade (*Khare & Lawrence, 2000*) to TLS encryption, as commonly used by Internet Printing Protocol for ipp URIs, but on the Web, best practice is explicit https (port 443) URLs to ensure following links stay secure.
** HATEOAS: Hypermedia as the Engine of Application State (*Fielding, 2000*), an important element of the REST architectural style.

- Many providers of FAIR Linked Data also provide programmatic REST API endpoints, *e.g.*, UNIPROT, ChEMBL, but keeping the FAIR aspects such as retrieving metadata in such a scenario may require combining different services using multiple formats and identifier conventions.

## Assessing FDO against FAIR

In addition to having "FAIR" in its name, the FAIR Digital Object guidelines (*Anders et al., 2023b*) also include *G3: FDOs must offer compliance with the FAIR principles through measurable indicators of FAIRness.*

In Table 5 we evaluate to what extent the FDO guidelines and its implementation with DOIP and Linked Data Platform (*Bonino da Silva Santos, Guizzardi & Sales, 2022*) comply with the FAIR principles (*Wilkinson et al., 2016*). Here we have used the RDA's FAIR Data Maturity Model (*FAIR Data Maturity Model Working Group, 2020*) as it has decomposed the FAIR principles to a structured list of FAIR indicators (*Bahim et al., 2020*), importantly considering *Data* and *Metadata* separately. In our interpretations we have, for simplicity, chosen to interpret "data" in FDOs as the associated bytestream of arbitrary formats, with remaining JSON or RDF structures always considered as metadata.

### Observations

- Linked Data in general is strong on metadata indicators, but LDP approach is weak as it has little concrete metadata guidance.
- FDO/DOIP are stronger on identifier indicators, while Linked Data approach for identifiers relies on best practices.
- Indicators on standard protocols (RDA-A1-04M, RDA-A1-04D, RDA-A1.1-01M, RDA-A1.1-01D) favour LDP's mature standards (HTTP, URI)—the DOIPv2 specification (*DOIPV2.0, 2018*) has currently only a couple of implementations and is expressed informally. The underlying Handle system for PIDs is arguably mature and

Soiland-Reyes et al. (2024), *PeerJ Comput. Sci.*, DOI 10.7717/peerj-cs.1781

**Table 5 Assessing RDA's FAIR Data Maturity Model (*FAIR Data Maturity Model Working Group, 2020*; *Bahim et al., 2020*) (first 2 columns) against the FDO guidelines (*Bonino et al., 2019*), FDO implemented with the protocol DOIPv2 (*DOIPV2.0, 2018*), Linked Data Platform (LDP) (*Bonino da Silva Santos, Guizzardi & Sales, 2022*) and examples from Linked Data practices in general.**

| FAIR ID | Indicator | FDO guidelines | FDO/DOIP | FDO/LDP | Linked Data examples |
|---|---|---|---|---|---|
| RDA-F1-01M | Metadata is identified by a persistent identifier | FDOF4 | Optional *Metadata FDO* w/separate PID | Content-negotiation to URL, not required to be PID | Metadata typically don't have own PID |
| RDA-F1-01D | Data is identified by a persistent identifier | FDOF1 | PIDs required (FDOF1). Handle, DOI. | FDOF-IR (Identifier Record). PID can be any URI | "Cool" URIs (*Berners-Lee, 1998*), PURL services incl. `purl.org`, `w3id.org` |
| RDA-F1-02M | Metadata is identified by a globally unique identifier | FDOF4 FDOF8 | Optional *Metadata FDO*, unspecified how to indicate | Content-negotiation to URL | Not required, content-negotiation can redirect to URL or `Content-Location`. FAIR Signposting. |
| RDA-F1-02D | Data is identified by a globally unique identifier | FDOF1 | All FDOs have PIDs (FDOF1), DOIP uses Handle system | FDOF-IR (Identifier Record) | Always accessed by URL |
| RDA-F2-01M | Rich metadata is provided to allow discovery | FDOF2 FDOF4 FDOF8 FDOF9 | FDO has key-value metadata. Unclear how to link to additional metadata. | FDOF-IR links to multiple metadata records | RDF-based metadata by content negotiation or FAIR Signposting. Embedded in landing page (RDFa). |
| RDA-F3-01M | Metadata includes the identifier for the data | — | `id` and `type` are required metadata elements PIDs, also implicit as requests must use PID | PID only required in FDOF-IR record. | PID inclusion typical, but often inconsistent (*e.g.*, `www.example.com` *vs* `example.com`) or missing (use of `<>` as *this* subject) |
| RDA-F4-01M | Metadata is offered in such a way that it can be harvested and indexed | FDOF10 | No, registries not required (except Data Type Registries). Handle registry only searchable by PID. | — | Not specified, several registries/catalogues for vocabularies/types (*e.g.*, (*NCBO BioPortal, 2022*)). Indexing by search engines if exposing HTML w/schema.org. |
| RDA-A1-01M | Metadata contains information to enable the user to get access to the data | FDOF3 FDOF6 | Directly by DOIP, but not included in FDO metadata. `handle.net` HTTP resolution may redirect to landing page | Any property can point to URIs, but unclear if it is data | Common with clickable "follow your nose" URLs |
| RDA-A1-02M | Metadata can be accessed manually (*i.e.*, with human intervention) | — | (Cordra HTML landing page from `handle.net` URIs) | Optional content-negotiation, *e.g.*, by Apache Marmotta, OpenLink Virtuoso | HTTP content-negotiation to HTML is common |
| RDA-A1-02D | Data can be accessed manually (*i.e.*, with human intervention) | — | (Cordra HTML landing page from `handle.net` URIs) | Optional content-negotiation | Direct download, HTML landing pages common for DOIs |

*(Continued)*

Soiland-Reyes et al. (2024), *PeerJ Comput. Sci.*, DOI 10.7717/peerj-cs.1781

| Table 5 (continued) | | | | |
|---|---|---|---|---|
| **FAIR ID** | **Indicator** | **FDO guidelines** | **FDO/DOIP** | **FDO/LDP** | **Linked Data examples** |
|---|---|---|---|---|---|
| RDA-A1-03M | Metadata identifier resolves to a metadata record | FDOF8+FDOF2 | — | — | Content-Location or HTTP redirection may indicate metadata URI |
| RDA-A1-03D | Data identifier resolves to a digital object | FDOF2 | Required, but frequently not directly resolvable | Recommended, but any URI acceptable | Resolvable HTTP/HTTPS URIs are most common, now infrequent URNs are not directly resolvable |
| RDA-A1-04M | Metadata is accessed through standardised protocol | G9 FDOF3 | Retrievable from PID (FDOF3). Informal DOIP standard maintained by DONA Foundation | LDP standard maintained by W3C, HTTP standards maintained by IETF, FDO components resolved by informal proposals (custom vocabulary, extra HTTP methods) or HTTP content negotiation) | Formal HTTP standards maintained by IETF, HTTP content negotiation, informal FAIR Signposting |
| RDA-A1-04D | Data is accessible through standardised protocol | G9 | (see above) | HTTP (*Fielding, Nottingham & Reschke, 2022*) | HTTP/HTTPS, FTP (now less common), GridFTP (*Allcock et al., 2005*) (for large data), ARK (*Kunze & Bermès, 2022*) |
| RDA-A1-05D | Data can be accessed automatically (*i.e.*, by a computer program) | G4 FDOF3 FDOF6 | Required, but few client libraries | HTTP GET, content-negotiation for fdof/object | Ubiquitous, hundreds of HTTP libraries |
| RDA-A1.1-01M | Metadata is accessible through a free access protocol | G1 G8 G9 | Partially realised: Handle system is open* protocol (*Sun et al., 2003*). One server implementation (*CNRI, 2022*), free. One DOIPv2 implementation (*Cordra*): free under BSD-like license (not recognised as Open Source). | LDP is open W3C recommendation (*Speicher, Arwe & Malhotra, 2015*). Multiple LDP implementations. | DNS, HTTP, TLS, RDF standards are open, free and universal, large number of Open Source clients and servers. |
| RDA-A1.1-01D | Data is accessible through a free access protocol | G9 | (see above) | URI, DNS, HTTP, TLS | URI, DNS, HTTP, TLS. Non-free DRM may be used (*e.g.*, subscription video streaming) |
| RDA-A1.2-01D | Data is accessible through an access protocol that supports authentication and authorisation | (FDOF9) | TLS certificates, authentication field (details unspecified) | Implied | HTTP authentication, TLS certificates |
| RDA-A2-01M | Metadata is guaranteed to remain available after data is no longer available | FDOF12 | — | Unspecified, however FDOF-IR links to separate metadata records | — |

| FAIR ID | Indicator | FDO guidelines | FDO/DOIP | FDO/LDP | Linked Data examples |
|---|---|---|---|---|---|
| RDA-I1-01M | Metadata uses knowledge representation expressed in standardised format | FDOF8 | Required, but not currently defined | — | Always implied by use of RDF syntaxes. |
| RDA-I1-01D | Data uses knowledge representation expressed in standardised format | — | — | — | Common (*e.g.*, HDF5, JSON, XML), yet common scientific data formats frequently not standardised |
| RDA-I1-02M | Metadata uses machine-understandable knowledge representation | FDOF8 | Required | Optional RDF metadata with any vocabulary | Always implied by use of RDF syntaxes. |
| RDA-I1-02D | Data uses machine-understandable knowledge representation | G4 G7 FDOF2 | No requirements on binary data formats | Only indirectly, LDP Basic Container reference only information resources | Common, specially for scientific data formats |
| RDA-I2-01M | Metadata uses FAIR-compliant vocabularies | G3 FDOF10 | Informally required | Unspecified, implied by use of RDF? | FAIR practices for LD vocabularies increasingly common, sometimes inconsistent (*e.g.*, PURLs that don't resolve) or incomplete (*e.g.*, unknown license) |
| RDA-I2-01D | Data uses FAIR-compliant vocabularies | — | — | — | Uncommon, except for some XML and RDF-embedding formats, *e.g.*, Extensible Metadata Platform (XMP) (*ISO 16684-1, 2019*) |
| RDA-I3-01M | Metadata includes references to other metadata | FDOF8 | Implied (attributes to PIDs), currently unspecified if given attribute is value or reference | — | By definition (Linked Data reference existing URIs (*W3C, 2015*)), `rdfs:seeAlso`, FAIR signposting (*Van de Sompel et al., 2022*) `describedby` |
| RDA-I3-01D | Data includes references to other data | G6 FDOF3 FDOF11 | — | — | URL hyperlinks common in several formats (HTML, PDF, JSON, XML). |
| RDA-I3-02M | Metadata includes references to other data | G6 FDOF3 FDOF8 | Implied from custom FDO type's attribute | LDP Direct Container members can be any resources | URI objects are frequently data references, may be indirect *via* PID |

(Continued)

Soiland-Reyes et al. (2024), *PeerJ Comput. Sci.*, DOI 10.7717/peerj-cs.1781

| FAIR ID | Indicator | FDO guidelines | FDO/DOIP | FDO/LDP | Linked Data examples |
|---|---|---|---|---|---|
| RDA-I3-02D | Data includes qualified references to other data | FDOF3 FDOF11 | Only indirectly through FDO metadata | Indirectly through LDP membership | Uncommon: Link relations, FAIR Signposting |
| RDA-I3-03M | Metadata includes qualified references to other metadata | (FDOF3) | Qualification by attribute keys defined per FDO Type | LDP Direct Container | Qualifications by property, PROV bundles (*Lebo & Moreau, 2013*), schema.org/Role |
| RDA-I3-04M | Metadata include qualified references to other data | (FDOF3) | Qualification by attribute keys defined per FDO type | LDP Indirect Container | Qualifications by property, n-ary indirection (schema.org Role (*Holland & Johnson, 2014*), `prov:specializationOf` (*Lebo, McGuinness & Sahoo, 2013*), OAI-ORE Proxy (*Lagoze et al., 2008*)) |
| RDA-R1-01M | Plurality of accurate and relevant attributes are provided to allow reuse | FDOF4 | Required. Kernel metadata attributes desired (*Broeder et al., 2022*) but not assigned PIDs yet. | Unspecified. Multiple metadata records can allow multiple semantic profiles. | Large number of general and domain-specific vocabularies can make it hard to find relevant attributes. Rough consensus on kernel metadata: schema.org (Schema.Org - *Schema.org, 2022a*), Dublin Core Terms (*DCMI Usage Board, 2020*), DCAT (*Browning et al., 2020*), FOAF (*Brickley & Miller, 2014*) |
| RDA-R1.1-01M | Metadata includes information about the licence under which the data can be reused | — | `licenseConditions` URL/PID in kernel metadata (*Broeder et al., 2022*) | — | Dublin Core Terms `dct:license` frequently recommended, frequently not required, *e.g.*, by DCAT 2 (*Browning et al., 2020*) |
| RDA-R1.1-02M | Metadata refers to a standard reuse licence | — | — | — | SPDX and Creative Commons URIs common, identifiers often inconsistent |
| RDA-R1.1-03M | Metadata refers to a machine-understandable reuse licence | — | — | — | SPDX documents uncommon |
| RDA-R1.2-01M | Metadata includes provenance information according to community-specific standards | FDOF9 FDOF10 | Unspecified (some Cordra types add getProvenance methods). PID Kernel attributes? | — | W3C PROV-O, PAV |

| FAIR ID | Indicator | FDO guidelines | FDO/DOIP | FDO/LDP | Linked Data examples |
|---------|-----------|----------------|----------|---------|----------------------|
| RDA-R1.2-02M | Metadata includes provenance information according to a cross-community language | FDOF9 FDOF8 | — | — | W3C PROV-O (*Lebo, McGuinness & Sahoo, 2013*), PAV (*Ciccarese et al., 2013*), Dublin Core Terms (*DCMI Usage Board, 2020*) |
| RDA-R1.3-01M | Metadata complies with a community standard | FDOF10 FROR8 | (Emerging, *e.g.*, DiSSCo Digital Specimen (*Hardisty et al., 2022*)) | — | Common, *e.g.*, DCAT 2 (*Browning et al., 2020*), BioSchemas (*Gray et al., 2017*) |
| RDA-R1.3-01D | Data complies with a community standard | (FDOF3) | — | — | Common, HTTP use registered IANA media types, additional scientific file formats frequently not standardised or identified |
| RDA-R1.3-02M | Metadata is expressed in compliance with a machine-understandable community standard | FDOF4 FDOF10 | Recommended | — | Common practice for ontologies, specially in bioinformatics, *e.g.*, BioPortal (*NCBO BioPortal, 2022*), Darwin Core (*Wieczorek et al., 2012*) |
| RDA-R1.3-02D | Data is expressed in compliance with a machine-understandable community standard | (FDOF2) | No, FDO is typed but data can be any bytestream | — | Occassionally, (*e.g.*, GFF3, FITS, ESRI) |

**Note:**

[*] The Handle.net system was previously covered by software patent US6135646A which expired in 2013.

[**] The Handle.net public license is not OSI-approved (*Open Source Initiative (OSI), 2022*) as an open source license—it includes usage restrictions and requires Service Agreements. It is not a DOIP requirement to host a local Handle instance, *e.g.*, EOSC provides the B2HANDLE service for acquiring Handle prefixes.

commonly used by researchers (this article alone references about 80 DOIs), however DOIs are more commonly accessed as HTTP redirects through resolvers like https://doi.org/ and http://hdl.handle.net/ rather than the Handle protocol.

- RDA-A1-02M and RDA-A1-02D highlights access by manual intervention, which is common for `http/https` URIs, but also using above PID resolvers for DOIP implementation CORDRA (*e.g.*, https://hdl.handle.net/21.14100/90ec1c7b-6f5e-4e12-9137-0cedd16d1bce), yet neither LDP, FDO nor DOIP specifications recommends human-readable representations to be provided
- Neither DOIP nor LDP require license to be expressed (RDA-R1.1-01M, RDA-R1.1-02M, RDA-R1.1-03M), yet this is crucial for re-use and machine actionability of FAIR data and metadata to be legal
- Machine-understandable types, provenance and data/metadata standards (RDA-R1.1-03M RDA-R1.3-02M, RDA-R1.3-02M, RDA-R1.3-02D) are important for machine actionability, but are currently unspecified for FDOs. *Blanchi et al. (2023)* explores possible machine-readable FDO types, however the type systems themselves have not yet been formalised. Linked Data on the other side have too many semantic and syntactic type systems, making it difficult to write consistent clients.
- Indicators for FAIR data are weak for either approach, as too much reliance is put on metadata. For instance in Linked Data, given a URL of a CSV file, what is its persistant identifier or license information? Signposting (*Van de Sompel & Nelson, 2015*) can improve findability of metadata using HTTP Link relations, which enable an FDO-like overlay for any HTTP resource. In DOIP, responses for bytestreams can include the data identifier: if that is a PID (not enforced by DOIP), its metadata is accessible.
- Resolving FDOs *via* Handle PIDs to the corresponding DOIP server is currently undefined by FDO and DOIP specifications. `0.TYPE/DOIPServiceInfo` lookup is only possible once DOIP server is known.

## EOSC interoperability framework

The European Open Science Cloud (EOSC) is a large EU initiative to promote Open Science by implementing a joint research infrastructure by federating existing and new services and focusing on interoperability, accessability, best practices as well as technical infrastructure (*Ayris et al., 2016*). The EOSC Interoperability Framework (*Corcho et al., 2021*) details the principles for creating a common way to achieve interoperability between all digital aspects of research activities in EOSC, including data, protocols and software. The recommendations are realized through four layers, Technical (*e.g.*, protocols), Semantic (*e.g.*, metadata models), Organisational (*e.g.*, recommendations) and Legal (*e.g.*, agreements), with a particular aim to address the FAIR interoperability principles and building on the concept of FAIR Digital Objects.

As covered in our introduction, EOSC proposes FAIR Digital Objects as a way to improve interoperability, for instance invoked by scientific workflows, carried by metadata frameworks and semantic artefacts. Therefore we here find it important to summarize how FDO and Linked Data can help satisfy the EOSC requirements.

**Table 6 Assessing EOSC Interoperability Framework (*Corcho et al., 2021*, section 3.6) against the FDO guidelines (*Bonino et al., 2019*) and Linked Data practices.**

| Layer | Recommendation | FDO | Linked Data |
|---|---|---|---|
| Technical | Open Specification | FDO specifications are semi-open, process gradually more transparent | Open and transparent standard processes through W3C & IETF |
| Technical | Common security & privacy framework | Unspecified | TLS for encryption, multiple approaches for single-sign-on (*e.g.*, ORCID, Life Science Login). Privacy largely unspecified. |
| Technical | Easy SLAs for service providers | Unspecified | None |
| Technical | Access data in different formats | None formalised, custom operations or relations | Content-negotiation, `rel=alternate` relations |
| Technical | Coarse-grained/fine-grained search tools | Freetext `0.DOIP/Op.Search` on local DOIP, no federation | Coarse-grained *e.g.*, Google Dataset Search, fine-grained (*e.g.*, federated SPARQL) require detailed vocabulary/metadata insight |
| Technical | Clear PID policy | Strong FDO requirements, tends towards Handle system. | Not required, different communities set policies |
| Semantic | Clear definitions for concepts/metadata/schemas | Required by FDO requirements, but not yet formalised | Ontologies, SKOS, OWL |
| Semantic | Semantic artefacts w/open licenses | All artefacts are PIDs, license not yet required by kernel metadata | Open License is best practice for ontology publishing |
| Semantic | Documentation for each semantic artefact | No direct rendering from FDO, no requirement for human-readable description | Ontology rendering, content-negotiation |
| Semantic | Repositories of artefacts | Required, but not formalised | Bioontologies, otherwise not usually federated |
| Semantic | Repositories w/clear governance | Recommended | Largely self-governed repositories, if well-established may have clear governance. |
| Semantic | Minimal metadata model for federated discovery | Kernel metadata (*Broeder et al., 2022*) based on RDA recommendations (*Weigel et al., 2018*). | DCAT, schema.org, Dublin Core |
| Semantic | Crosswalks from minimal metadata model | FDO Typing recommends referencing existing type definitions, but not as separate crosswalks | Multiple crosswalks for common metadata models, but frequently not in semantic format |
| Semantic | Extensibility options for diciplinary metadata | Communities encouraged to establish own types | Extensible by design, domain-specific metadata may be at different granularity |
| Semantic | Clear protocols/building blocks for federation/harvesting of artefact catalogues | Collection types not yet defined | SWORD, OAI-PMH |
| Organisational | Interoperability-focused rules of participation recommendations | Recommended | Implied only by some communities, tendency to specialise |
| Organisational | Usage recommendations of standardised data formats | None | None—but common for metadata (*e.g.*, JSON-LD) |
| Organisational | Usage recommendations of vocabularies | Recommended by community | Common (see RDMKit) |
| Organisational | Usage recommendations of metadata | Recommended by community | RO-Crate, Bioschemas |
| Organisational | Management of permanent organization names/functions | Handle owner, but unclear contact. Contact info in DOIP service provider | ROR. DCAT contacts. |
| Legal | Standardised human and machine-readable licenses | None | SPDX, but not that frequently used |

(Continued)

| Layer | Recommendation | FDO | Linked Data |
|-------|----------------|-----|-------------|
| Legal | Permissive licenses for metadata (CC0, CC-BY-4.0) | Undefined | Both CC0, CC-BY-4.0 common, *e.g.*, in DCAT |
| Legal | Different licenses for different parts | Each part as separate FDO can have separate license | DCAT, RO-Crate, Named graphs for splitting metadata |
| Legal | Mark expired/inexistent copyright | Undefined | Unclear, semantics assume copyright valid |
| Legal | Mark orphaned data | Tombstone for deleted data, but no owner of DOIP server means FDO disappears | Frequently data and endpoint has no known maintainer, archiving in common repositories becoming common |
| Legal | List recommended licenses | Undefined | Best practice recommendations |
| Legal | Track license evolution for dataset | Undefined | Versioning with PAV/PROV/DCAT |
| Legal | Policy/guidance for patent/trade secrets violation | Undefined | Undefined, legal owner may be specified. ODRL can express policies. |
| Legal | GDPR compliance for personal data | Undefined | Undefined |
| Legal | Restrict access/use if legally required | By transport protocol (undefined by FDO/DOIP) | Diverging approaches, typically landing pages w/ auth&auth or click-thru |
| Legal | Harmonised terms-of-use | Undefined | Undefined |
| Legal | Alignment between EOSC and national legislation | Not applicable | Not applicable |

In Table 6 we review the EOSC Interoperability Framework (EOSC IF) recommendations, and evaluate to what extent they are addressed by the principles of FDO and Linked Data or their common implementations.

### Observations

Firstly, we observe that the EOSC IF recommendations are at a high level, mainly affecting governance and practices by communities. This *Organizational* level is also highlighted by the FDO recommendations, for instance the FDO Typing (*Lannom et al., 2022c*) propose a governance structure to recognize community-endorsed services. While these community aspects are not mandated by Linked Data practices, best practices have become established for aspects like ontology development (*Norris et al., 2021*). EOSC IF's *Technical* layer is likewise at a architecturally high level, such as service-level agreements, but also highlight PID policies which is strongly required by FDO, while Linked Data communities choose PID practices separately. The recommendations for the *Semantic* layer is largely already implemented by Linked Data practices, yet for FDO mostly consist of encouragements. For instance *clear definitions of semantic concepts* is required by FDO guidelines, but how to technically define them has not been formalised by FDO specifications.

The *Legal* layer of interoperability is perhaps the one most emphasised by EOSC, by enabling collaboration across organizational barriers to joinly build a research infrastructure, but this is an area that both FDO and Linked Data are relatively weak in directly supporting. The EOSC IF recommendations in this layer are largely related to

governance practices and metadata, for instance licensing, privacy and usage policies; these are also essential for cross-institutional and cross-repository access of FAIR objects.

Likewise, search and indexing is important FAIR aspect for Findability, but is poorly supported globally by FDO and Linked Data. Efforts such as Open Research Knowledge Graph (ORKG) (*Jaradeh et al., 2019*), DataCite's PID Graph (*Fenner & Aryani, 2019*) and Google Knowledge Graph (*Singhal, 2012*) have improved programmatic findability to some degree, however not significantly for domain-specific semantic artefacts, currently scattered across multiple semantic catalogues (*Corcho et al., 2023*). There is a strong role for organizations like EOSC to provide such broader registries, moving beyond scholarly output metadata federations. The EOSC Marketplace (https://marketplace.eosc-portal.eu/) has for instance recently been expanded to include training material, software and data sources.

## DISCUSSION

We have evaluated the FAIR Digital Object concept using multiple frameworks, and contrasted FDO against existing experiences from Linked Data on the Web. In this section we discuss the implications of this evaluation, and propose how these two approaches can be better combined.

### Framework evaluation

Having considered FDO and the Web architecture as interoperability frameworks we observe that neither are magic bullets, but each bring different aspects of interoperability. The Web comes with a large degree of flexibility and openness, however this means interoperability can suffer as services have different APIs and data models, although with common patterns. This is also true for Linked Data on the Web, with many overlapping ontologies and frequent inconsistencies in resolution mechanisms; although somewhat alleviated in recent years by schema.org becoming common metadata model for semantic markup inline in Web pages. The Web is based on a common HTTP protocol which has remained stable architecturally throughout its 32 years of largely backwards-compatible evolution. FDO on the other side sets down multiple rigid rules for identifiers, types, methods *etc.* that are adventurous for interoperability and predictability for FAIR consumption. Yet there is a large degree of freedom in how the FDO rules can be implemented by a given community, for instance there is no common metadata model or identifier resolution mechanism, and DOIP is just one possible transport method for FDOs, which itself does not enforce these rules.

When evaluating FDO implementations against the FDO guidelines we see that several technical pieces and community practices still need to be developed and further defined, for instance the FDO type system, how to declare FDO actions, how to resolve persistent identifiers, or how to know which pattern of FDO composition is used. Achieving fully interoperable FAIR Digital Objects would require further convergence on implementation practices, and it is not given that this needs to diverge from the established Web architecture. It is not clear from FDO guidelines if moving from HTTP/DNS to DOIP/ Handle as a way to expose distributed digital objects will benefit FAIR practitioners, when

both approaches require additional equally implementable restrictions and conventions, such as using persistent identifiers or pre-defining an object's type.

Considering this, by comparing FDO and Web as middleware we saw that programmatic access to digital objects, a core promise of FDO, is not particularly improved by the use of the protocol DOIP as compared to HTTP, *e.g.*, lack of concurrency and transparency. Recent updates to HTTP have added many features needed for large-scale usage such as video streaming services (*e.g.*, caching, multiplexing, cloud deployments), and having the option to transparently apply these also to FDOs seems like a strong incentive. Many programmatic features for distributed objects are however missing or needing custom extensions in both aspects, such as transactions, asynchronous operations and streaming.

By assessing FDO against the FAIR principles we found that both FDO implementations are underspecified in several aspects (licences, provenance, data references, data vocabularies, metadata persistence). While there are implementations of each of these in general Linked Data examples, there is no single set of implementation guides that fully realizes the FAIR principles. *FAIRification* efforts like the FAIR Cookbook (*Rocca-Serra et al., 2023*) and FAIR Implementation Profiles (*Schultes et al., 2020*) are bringing existing practices together, but there remains a potential role for FDO in giving a coherent set of implementation practices that can practically achieve FAIR. Significant effort, also within EOSC, is now moving towards FAIR metrics (*Devaraju et al., 2021*), which in practice need to make additional assumptions on how FAIR principles are implemented, but these are not always formalized (*Wilkinson et al., 2022b*) nor can they be taken to be universally correct (*Verburg et al., 2023*). Given that most of the existing FAIR guides and assessment tools are focused on Web and Linked Data, it would be reasonable for FDO to then provide a profile of such implementation choices that can achieve best of both worlds.

EOSC has been largely supportive of FDO, FAIR and related services. By contrasting the EOSC Interoperability Framework with FDO, we found that there are important dimensions that are not solved at a technical level, but through organization collaboration, legal requirements and building community practices. FDO recommendations highlight community aspects, but at the same time the largest FAIR communities in many science domains are already producing and consuming Linked Data. Just as the Linked Data community has a challenge in convincing more research fields to use Semantic Web technologies, FDO currently need to build many new communities in areas that have shown interest in that approach (*e.g.*, material science). It may be advantageous for both these effort to be aligned and jointly promoted under the EOSC umbrella.

## What does FDO mean for Linked Data?

The FAIR Digital Object approach raises many important points for Linked Data practitioners. At first glance, the explicit requirements of FDOs may seem to be easy to furfill by different parts of the Semantic Web Cake (*Berners-Lee, 2000*, slide 10), as has previously been proposed (*Soiland-Reyes et al., 2022a*). However, this deeper investigation,

based on multiple frameworks, highlights that the openness and variability of how Linked Data is deployed can make it difficult to achieve the FDO goals without significant effort.

While RDF and Linked Data have been suggested as prime candidates for making FAIR data, we argue that when different developers have too many degrees of freedom (such as serialization formats, vocabularies, identifiers, navigation), interoperability is hampered—this makes it hard for machines to reliably consume multiple FAIR resources across repositories and data providers. Indeed, this may be one reason why the initial FDO effort steered away from Linked Data approaches, but now seems in a danger of opening the many same degrees of freedom within FDO.

We therefore identify the need for a new explicit FDO profile of Linked Data that sets pragmatic constraints and stronger recommendations for consistent and developer-friendly deployment of digital objects. Such a combination of efforts could utillise both the benefits of mature Semantic Web technologies (*e.g.*, federated knowledge graph queries and rich validation) and data management practices that follow FDO guidance in order to grow an ecosystem of machine-actionable objects. It is beyond the scope of this work to detail such a profile, but we suggest the following potential key aspects:

- Use HTTP(S) as protocol
- Use URIs as identifiers, with persistent identifier promises
- Provide consistent identifier resolution that does not require heuristics
- Common core metadata model
- References are always URIs, and should be persistent identifiers
- Types, attributes and actions are self-defined by their identifier
- Use Web approaches directly where possible, rather than wrap in a new model

The FAIR and Linked Data communities likewise need to recognize the need for simpler, more pragmatic approaches that make it easier for FAIR practitioners to adapt the technologies with "just enough" semantics.

## CONCLUSION

In this work, we have considered FAIR Digital Objects (FDO) as a potential distributed object system for FAIR data and compared it with established Web approaches focusing on Linked Data. We have described the background of the Semantic Web and FAIR Digital Objects, and evaluated both using multiple conceptual frameworks.

We find that both FDO and Linked Data approaches can significantly benefit from each-other and should be aligned further. Namely, Linked Data proponents need to make their technologies more approachable, agreeing on predictable and consistent implementations of FAIR principles.

The FDO recommendations show that FAIR thinking in this regard need to move beyond data publishing and into machine actionability across digital objects, and with broader community consensus. As flexibility for extensions is a necessary ingredient alongside rigidity for core concepts, the FDO community likewise need to settle on directly

implementable specifications rather than just guidelines, and avoid making similar mistakes as learnt by early Semantic Web adopters.

By implementing the goals of FAIR Digital Objects with the mature technology stack developed for Linked Data, EOSC research infrastructures and researchers in general can create and use FAIR machine-actionable research outputs for decades to come.

## AN OVERVIEW OF UPCOMING FDO SPECIFICATIONS

**FAIR Digital Object Overview and Specifications** (*Anders et al., 2023a*) is a comprehensive overview of FAIR Digital Object specifications listed below. It serves as a primer that introduces FDO concepts and the remaining documents. It is accompanied by an FDO Glossary (*Broeder & Wittenburg, 2022*).

The **FDO Forum Document Standards** (*Weiland et al., 2022b*) documents the recommendation process within the forum, starting at *Working Draft* (WD) status within the closed working group and later within the open forum, then *Proposed Recommendation* (PR) published for public review, finalised as *FDO Forum Recommendation* (REC) following any revisions. In addition, the forum may choose to *endorse* existing third-party notes and specifications.

The **FDO Requirement Specifications** (*Anders et al., 2023b*) is an update of *Bonino et al. (2019)* as the foundational definition of FDO. This sets the criteria for classifying an digital entity as a FAIR Digital Object, allowing for multiple implementations. The requirements shown in Table 3 are largely equivalent, but in this specification clarified with references to other FDO documents.

**Machine Actionability** (*Weiland et al., 2022a*) sets out to define what is meant by *machine actionability* for FDOs. *Machine readable* is defined as elements of bit-sequences defined by structural specification, *machine interpretable* elements that can be identified and related with semantic artefacts, while *machine actionable* are elements with a type with operations in a symbolic grammar. The document largely describes requirements for resolving an FDO to metadata, and how types should be related to possible operations.

**Configuration Types** (*Lannom et al., 2022a*) classifies different granularities for organising FDOs in terms of PIDs, PID Records, Metadata and bit sequences, *e.g.*, as a single FDO or several daisy-chained FDOs. Different patterns used by current DOIP deployments are considered, as well as FAIR Signposting (*Van de Sompel & Nelson, 2015*; *Van de Sompel et al., 2022*).

**PID Profiles & Attributes** (*Anders et al., 2022*) specifies that PIDs must be formally associated with a *PID Profile*, a separate FDO that defines attributes required and recommended by FDOs following said profile. This forms the *kernel attributes*, building on recommendations from RDA's *PID Information Types* working group (*Weigel et al., 2018*). This document makes a clear distinction between a minimal set of attributes needed for PID resolution and FDO navigation, which needs to be part of the *PID Record* (*Sharif, 2023*), compared with a richer set of more specific attributes as part of the *metadata* for an FDO, possibly represented as a separate FDO.

**Kernel Attributes & Metadata** (*Broeder et al., 2022*) elaborates on categories of FDO Mandatory, FDO Optional and Community Attributes, recommending kernel attributes

like `dateCreated`, `ScientificDomain`, `PersistencePolicy`, `digitalObjectMutability`, *etc*. This document expands on RDA Recommendation on PID Kernel Information (*Weigel et al., 2018*). It is worth noting that both documents are relatively abstract and do not establish PIDs or namespaces for the kernel attributes.

**zenodo.8075229 has been applied** by several institutions (*Wittenburg et al., 2022*). The document proposes that DOIP shall be assessed for completeness against FDO—in this initial draft this is justified as "*we can state that DOIP is compliant with the FDO specification documents in process*" (the documents listed above).

**Granularity, Versioning, Mutability** (*Hellström, Zwölf & Wittenburg, 2022*) considers how granularity decisions for forming FDOs must be agreed by different communities depending on their pragmatic usage requirements. The affect on versioning, mutability and changes to PIDs are considered, based on use cases and existing PID practices.

**DOIP Endorsement Request** (*Schwardmann et al., 2022*) is an endorsement of the DOIP v2.0 (DONA 2018) specification as a potential FDO implementation, as it has been applied by several institutions (*Wittenburg et al., 2022*). The document proposes that DOIP shall be assessed for completeness against FDO–in this initial draft this is justified as "*we can state that DOIP is compliant with the FDO specification documents in process*" (the documents listed above).

**Upload of FDO** (*Blanchi et al., 2022*) illustrates the operations for uploading an FDO to a repository, what checks it should do (for instance conformance with the PID Profile, if PIDs resolve). ResourceSync (*ANSIZ39.99, 2017*) is suggested as one type of service to list FDOs. This document highlights potential practices by repositories and their clients, without adding any particular requirements.

**Typing FAIR Digital Objects** (*Lannom et al., 2022c*) defines what *type* means for FDOs, primarily to enable machine actionability and to define an FDO's purpose. This document lays out requirements for how *FDO Types* should themselves be specified as FDOs, and how an *FDO Type Framework* allows organising and locating types. Operations applicable to an FDO is not predefined for a type, however operations naturally will require certain FDO types to work. How to define such FDO operations is not specified.

**Implementation of Attributes, Types, Profiles and Registries** (*Blanchi et al., 2023*) details how to establish FDO registries for types and FDO profiles, with their association with PID systems. This document suggest policies and governance structures, together with guidelines for implementations, but without mandating any explicit technology choices. Differences in use of attributes are examplified using FDO PIDs for scientific instruments, and the proto-FDO approach of DARIAH-DE (*Schwardmann & Kálmán, 2022*).

See bibliography below for the citation per document above.

## FDO specifications

**FDO-Overview-PEN-2.0** Ivonne Anders, Christophe Blanchi, Daan Broder, Maggie Hellström, Sharif Islam, Thomas Jejkal, Larry Lannom, Karsten Peters-von Gehlen, Robert Quick, Alexander Schlemmer, Ulrich Schwardmann, Stian Soiland-Reyes, George Strawn, Dieter van Uytvanck, Claus Weiland, Peter Wittenburg, and Carlo Zwölf (2023-01-18).

*FAIR Digital Object Technical Overview*. Version PEN 2.0. Proposed Recommendation Full FDO Overview PEN-2.0-v2. https://doi.org/10.5281/zenodo.7824714.

**PED-DOIPEndorsement-1.1** Ulrich Schwardmann, George Strawn, Robert Quick, and Peter Wittenburg (2022-10-17). *DOIP Endorsement Request*. Enforcement Request PED-DOIPEndorsement-1.1-20221017. https://doi.org/10.5281/zenodo.7824796.

**PEN-FDO-Upload** Christophe Blanchi, Daan Broeder, Thomas Jejkal, Islam Sharif, Alexander Schlemmer, Dieter van Uytvanck, and Peter Wittenburg (2022-10-17). *FDO— Upload of FDO*. Proposed Endorsement Note PEN-FDO-Upload-1.1-20221017. https://doi.org/10.5281/zenodo.7825549.

**PR-ConfigurationTypes-2.1** Larry Lannom, Karsten Peters-von Gehlen, Ivonne Anders, Andreas Pfeil, Alexander Schlemmer, Zach Trautt, and PeterWittenburg (2022-10-17). *FDO Configuration Types*. Proposed Recommendation PR-ConfigurationTypes-2.1-20221017. https://doi.org/10.5281/zenodo.7825703.

**PR-KernelAttributues-2.0** Daan Broeder, Peter Wittenburg, Ivonne Anders, and Karsten Petersvon Gehlen (2022-10-17). *FDO—Kernel Attributes & Metadata*. Proposed Recommendation PR-FDO-KernelAttributesAndMetadata-2.0-20221017. https://doi.org/10.5281/zenodo.7825693.

**PR-MachineActionDef-2.2** Claus Weiland, Sharif Islam, Daan Broder, Ivonne Anders, and Peter Wittenburg (2022-11-19). *FDO Machine Actionability*. Version 2.2. Proposed Recommendation PR-MachineActionDef-2.2-20221119. https://doi.org/10.5281/zenodo.7825650.

**PR-PIDProfileAttributes-2.1** Ivonne Anders, Maggie Hellström, Sharif Islam, Thomas Jejkal, Larry Lannom, Ulrich Schwardmann, and Peter Wittenburg (2022-10-17). *FDO PID Profiles & Attributes*. Proposed Recommendation PR-PIDProfileAttributes-2.1-20221017. https://doi.org/10.5281/zenodo.7825630.

**PR-RequirementSpec-3.0** Ivonne Anders, Christophe Blanchi, Daan Broder, Maggie Hellström, Sharif Islam, Thomas Jejkal, Larry Lannom, Karsten Peters-von Gehlen, Robert Quick, Alexander Schlemmer, Ulrich Schwardmann, Stian Soiland-Reyes, George Strawn, Dieter van Uytvanck, Claus Weiland, Peter Wittenburg, and Carlo Zwoölf (2023-01-12). *FDO Forum FDO Requirement Specifications*. Version 3.0. Proposed Recommendation PR-RequirementSpec-3.0. https://doi.org/10.5281/zenodo.7782262.

**PR-TypingFDOs-2.0** Larry Lannom, Ulrich Schwardmann, Cristophe Blanchi, and Peter Wittenburg (2022-06-08). *Typing FAIR Digital Objects*. Proposed Recommendation PR-TypingFDOs-2.0-20220608. https://doi.org/10.5281/zenodo.7825599.

Weiland, C., Schwardmann, U., Wittenburg, P., Kirkpatrick, C., Hanisch, R., & Trautt, Z. (2022). FDO Forum Document Standards 1.1 (WD-DocProcessStd-1.1). FAIR Digital Object Forum. https://doi.org/10.5281/zenodo.1094337.

**WD-ImplAttributesTypesProfiles** Christophe Blanchi, Maggie Hellström, Larry Lannom, Andreas Pfeil, Ulrich Schwardmann, and Peter Wittenburg (2023-03-14). *Implementation of Attributes, Types, Profiles and Registries*. Working Draft WD-Implementation-of-Attributes-0.4-20230314. https://doi.org/10.5281/zenodo.7825572.

## ACKNOWLEDGEMENTS

We would like to acknowledge the FAIR Digital Object Forum (*FAIR Digital Objects, 2022a*) community and working groups, where SSR and CG are members.

Views and opinions expressed in this work are those of the authors only and do not necessarily reflect those of the funded projects, FAIR Digital Object Forum, European Union nor the European Commission.

### Funding

This work was funded by the European Union programmes Horizon 2020 under grant agreements H2020-INFRAEDI-02-2018 823830 (BioExcel-2), H2020-INFRAEOSC-2018-2824087 (EOSC-Life) and Horizon Europe under grant agreements HORIZON-INFRA-2021-EMERGENCY-01 101046203 (BY-COVID), HORIZON-INFRA-2021-EOSC-01 101057388 (EuroScienceGateway), HORIZON-INFRA-2021-EOSC-01-05 101057344 (FAIR-IMPACT), HORIZON-INFRA-2021-TECH-01 101057437 (BioDT); HORIZON-CL4-2021-HUMAN-01-01 101070305 (ENEXA) and by UK Research and Innovation (UKRI) under the UK government's Horizon Europe funding guarantee grants 10038963 (EuroScienceGateway), 10038992 (FAIR-IMPACT), 10038930 (BioDT). The funders had no role in study design, data collection and analysis, decision to publish, or preparation of the manuscript.

### Grant Disclosures

The following grant information was disclosed by the authors:
European Union programmes Horizon 2020: H2020-INFRAEDI-02-2018 823830 (BioExcel-2), H2020-INFRAEOSC-2018-2 824087 (EOSC-Life).
Horizon Europe: HORIZON-INFRA-2021-EMERGENCY-01 101046203 (BY-COVID), HORIZON-INFRA-2021-EOSC-01 101057388 (EuroScienceGateway), HORIZON-INFRA-2021-EOSC-01-05 101057344 (FAIR-IMPACT), HORIZON-INFRA-2021-TECH-01 101057437 (BioDT); HORIZON-CL4-2021-HUMAN-01-01 101070305 (ENEXA).
UK Research and Innovation (UKRI) under the UK government's Horizon Europe funding guarantee: 10038963 (EuroScienceGateway), 10038992 (FAIR-IMPACT), 10038930 (BioDT).

### Competing Interests

Stian Soiland-Reyes and Carole Goble are members of FAIR Digital Object forum with no financial compensation.

### Author Contributions

- Stian Soiland-Reyes conceived and designed the experiments, performed the experiments, analyzed the data, prepared figures and/or tables, authored or reviewed drafts of the article, conceptualization, data curation, formal analysis, funding

acquisition, investigation, methodology, writing-original draft, writing-review & editing, and approved the final draft.

- Carole Goble conceived and designed the experiments, authored or reviewed drafts of the article, funding acquisition, supervision, writing-review & editing, and approved the final draft.
- Paul Groth conceived and designed the experiments, authored or reviewed drafts of the article, conceptualization, methodology, supervision, writing-review & editing, and approved the final draft.

## Data Availability

The RO-Crate and the tables are available at Zenodo: (*Soiland-Reyes, 2023a*). https://doi.org/10.5281/zenodo.8075230.

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
