# Peer review of "Evaluating FAIR Digital Object and Linked Data as distributed object systems"

_PeerJ Computer Science, doi:10.7717/peerj-cs.1781_

## Round 0.1 · original submission · Minor Revisions

While both reviewers liked the work, they make several recommendations for improving your work. This includes in particular cleaning up issues with the writing in terms of both fixing typos as well as adding details and explanations. Please address these concerns.

**Language Note:** The Academic Editor has identified that the English language must be improved. PeerJ can provide language editing services - please contact us at [email protected] for pricing (be sure to provide your manuscript number and title). Alternatively, you should make your own arrangements to improve the language quality and provide details in your response letter. – PeerJ Staff

Reviewer 1 ·

Basic reporting

The article is written in English and use clear, unambiguous, technically correct text. The article must conform to professional standards of courtesy and expression. The only comment refers to “to form an ecosystem of rich digital objects” – it would be suggested to elaborate on what “rich” DO stands for, i.e. while this can be understood by the community, it should be suited for other communities as well, as well as it is always beneficial to set a common ground of understanding of the used terms.

The article includes sufficient introduction and background to demonstrate how the work fits into the broader field of knowledge. Relevant prior literature is mostly referenced. However, several comments are as follows:
1. It would be beneficial to conclude the Introduction with a brief section-based overview.
2. “The premise of systematically building an ecosystem of such digital objects is 70 to give researchers a way to organise complex digital entities, associated with identifiers, metadata, and 71 supporting automated processing” – while it is definitely true, yet another point to be mentioned here for both buildings and maintaining such ecosystems is governance. It would be expected to be mentioned here, esp. considering the recent report of EOSC, namely Wilkinson, M. D., Sansone, S. A., Méndez, E., David, R., Dennis, R., Hecker, D., ... & Castro, L. J. (2022). Community-driven governance of FAIRness assessment: an open issue, an open discussion. Open Research Europe, 2(146), 146.
3. At the same time according to the community, incl. EOSC emphasize that DO can be more than a single object with the reference to IS that can also be considered as DO and hence FAIR principles should be followed for them as well. This would be yet another point I would expect to see. See for instance (not limited to) - Azeroual, O., Schöpfel, J., Pölönen, J., & Nikiforova, A. (2022, October). Putting FAIR principles in the context of research information: FAIRness for CRIS and CRIS for FAIRness. In 14th International Conference on Knowledge Management and Information Systems (KMIS2022).
3. When referring to RDA, GO-FAIR, EOSC etc. I would suggest adding the relevant references for the sake of simplicity for the reader to access them easily – these could be footnotes

The structure of the article mostly conforms to an acceptable format of ‘standard sections’. Figures and Tables are relevant to the content of the article, of sufficient resolution, and appropriately described and labeled. However:
1. “Next steps for FDO” section could benefit of restructuring into a general overview of the further steps with the reference to those documents or just listing them instead of referring to each of them, which although contributes to the overall structure, at times provide a limited added value to the section since is more reflective on the actual aim of the working draft and a limited information about the actual next steps for FDO. Alternatively, you could divide them into those that contribute directly to the section purpose, and those that are rather descriptive and less valuable and leave those directly contributing in the form as-is, and reworking in accordance with the above suggestion those that do not.

The submission is ‘self-contained,’ represents an appropriate ‘unit of publication’, and includes all results relevant to the hypothesis.

Experimental design

This is an original primary research within Aims and Scope of the journal.
The submission defines the research question, which are relevant and meaningful - although they could be strengthen, i.e. presentation seems to be weaker compared to the actual content. The knowledge gap investigated is identified, and statements are made as to how the study contributes to filling that gap.

The investigation is conducted rigorously and to a high technical standard. However:
1. “As these documents clarify the future aims and focus of FAIR Digital Objects (Lannom, 100 Schwardmann, Christophe Blanchi, et al. 2022), we provide a brief summary of each” – please comment of how “each” was selected, i.e. whether they come from a single source? Or was this rather the result of the literature review or similar?
2. Table 3 – due to current deficiencies of the paper structure, referring the reader back and towards many times, it is not obvious what is the source of those guidelines aka requirements in the text in Table 3? Are they extracted from those documents reviewed before? Please specify and emphasize this clearly.
3. Table 3 evaluates completeness of the guidelines, however, while some detail regarding how this was conducted are provided, more detail would be expected on how this was actually done rather than referring the reader to external documents only. This would contribute to the replicability and reproducibility of the study. Also, whether the validity of both extraction of those guidelines/requirements and their assessment was conducted? How was this done? Or, if not, why it is not needed?

Validity of the findings

The contribution will be of interest for the audience / community

Sufficient amount and nature of the data have been provided or alternatively explicitly described

The conclusions are appropriately stated, connected to the original question investigated, and limited to those supported by the results.

Additional comments

Additionally, two minor comments:
1. “The FAIR principles (Mark D. Wilkinson et al. 2016) encourage sharing of scientific data…” – it is highly suggested to put the reference either at the end of the sentence, or, alternatively, rewording the sentence, e.g., as “the FAIR principles first defined by [REF]”. Please do so for all other occurrences.
2. “see Next steps for FDO on the following page” – instead of referring to another page, I would suggest referring the reader to the specific section. Please do so for other occurrences of the above as well.

Cite this review as

·

Basic reporting

The article uses a clear, unambiguous, professional English language throughout.
The context is shown by an introduction and background section.
The literature is well referenced and relevant.
The structure conforms to PeerJ standards,
The article does not contain any figures, which is not a drawback in this
case of a comparison of two technological approaches.
However it contains a couple of tables that are relevant, well labelled and described.
There is no raw data supplied.

Experimental design

This article provides original primary research within the scope of
the journal. The research question is well defined, relevant and meaningful.
It is stated how the research fills an identified knowledge gap.
It covers the interesting question how the FAIR Digital Object (FDO)
approach in comparison to the linked data approach fulfills the
requirements of the FAIR principles. This comparison is made along the lines
of five conceptual frameworks that capture different perspectives of the FAIR
principles, for instance stability, openness,
interoperability and readiness for implementation. Additionally different
existing implementation approaches and practices are taken into account
for FDOs and Linked Data.

The five conceptual frameworks are well chosen, because they cover a wide
range of aspects crucial for establishing an environment of FAIR data.
These frameworks are able to investigate and compare the different
dimensions of FAIR data like Data, Metadata, Service, Access,
Operations, Computation along the different viewpoints of stability and trustworthiness,
machine actionability or interoperability.

The methods are described with sufficient detail and can be easily replicated.
The five frameworks cover different levels of abstraction from highly
technological and conceptual aspects in the Interoperability Framework for
Fast Data to very concrete requirements for FAIRness in the RDA’s FAIR Data
Maturity Model that is a further specification of the FAIR Digital
Object guidelines. Partly the frameworks are strongly related to each other,
as the RDA’s FAIR Data Maturity Model and the FDO guidelines. This is
only very shortly mentioned in the Methods section.

For the reader a more precise description about the generality and
specificity of the frameworks wrt. FAIRness would be helpful in order to
get a better overview of the covered landscape. Whether additionally
a rearrangement of the
framework subsections in the Results section in an order of generality will
improve readability should be left to the authors.

The article gives clear statements about current lacks in the technology that
are partly due to the chosen implementation, but also partly due to missing
standardization or even conceptual ambiguity. These findings are valuable
criteria for both approaches in order to improve their FAIR readiness.

A missing aspect in the article is the role of profiles for FDOs. They are
shortly mentioned in the overview of detailed requirement documents in the
subsection on Next steps for FDOs. But their importance as the ubiquitous first
description of the syntactical and semantic structure of an FDO for
interoperability is not really covered in the article. Even if this conceptual
approach is not adopted by the FDO implementations under investigation, the
role of profiles should be mentioned at least in the subsection on Next steps
for FDOs, perhaps in the context of the 'PID Profiles & Attributes' or the
'Implementation of Attributes, Types, Profiles and Registries'
This should be referenced later in the conceptual reasoning of
the quality levels of the Interoperability Framework and when it comes to
suggestions regarding improvements in machine actionability and interoperability.

Validity of the findings

The conclusions are appropriately stated and connected to the
original question investigated. They are limited to those supported by
the results. The article is based on referenced publication. Raw data is not
supplied and not necessary here.

Additional comments

Stylistic Remarks

It would be advantageous for the reader to easily recognize the conclusions
drawn from the deeper analysis along each of the five frameworks, because these
conclusions are the main guidelines for improvements. They have to be read and
taken into account for the further development of the underlying technology
and existing implementations.

Such a better recognition and orientation could be reached by a consistent
format like a special subsection or for example the common use of punctuation
marks for all conclusions of the five frameworks.

Typos

line 57: ... has been introduced way as + a + way to ...

line 1123: PR-KernelAtributues-2.0 -> PR-KernelAttributes-2.0

line 131: also change the references at l131, at RDA-R1-01M in table5 p23, at 'Minimal metadata model for
federated discovery' in table6.

---

## Round 0.2 · accepted · Accept

I confirm that the authors have revised the document reasonably as per the minor revision notes suggested by the reviewers, and so the revised document is now acceptable.